**EMBO** *reports*

# Malaria parasites undergo a rapid and extensive metamorphosis after invasion of the host erythrocyte

Aline Fréville [1], Flavia Moreira-Leite[2], Camille Roussel[3], Matthew R G Russell [4,9], Aurelie Fricot[3], Valentine Carret [3], Abdoulaye Sissoko [3], Matthew J Hayes [5], Aissatou Bailo Diallo[3], Nicole Cristine Kerkhoven[3], Margarida Ressurreição[1], Safi Dokmak[6], Michael J Blackman [1,7], Lucy M Collinson [4], Pierre A Buffet[3], Sue Vaughan[2], Papa Alioune Ndour [3] & Christiaan van Ooij [1,8 ✉]

## Abstract

Within the human host, the symptoms of malaria are caused by the replication of malaria parasites within erythrocytes. Growth inside the erythrocyte exposes the parasites to the normal surveillance of erythrocytes by the host organism, in particular the clearance of erythrocytes in the spleen. Here we show that the malaria parasite *Plasmodium falciparum* undergoes a rapid, multi-step metamorphosis that transforms the invasive merozoite into an amoeboid-shaped cell within minutes after invading erythrocytes. This transformation involves an increase in the parasite surface area and is mediated by factors already present in the merozoite, including the parasite phospholipid transfer protein PV6. Parasites lacking PV6 do not assume an amoeboid form and instead are spherical and have a smaller surface area than amoeboid forms. Furthermore, erythrocytes infected with *P. falciparum* parasites lacking PV6 undergo a higher loss of surface area upon infection, which affects the traversal of infected erythrocytes through the spleen. This is the first evidence that after invasion, the parasite undergoes a rapid, complex metamorphosis within the host erythrocyte that promotes survival in the host.

**Keywords** Malaria; *Plasmodium*; Host-pathogen Interaction; Membranes
**Subject Categories** Membranes & Trafficking; Microbiology, Virology & Host Pathogen Interaction

## Introduction

Malaria caused by the parasite *Plasmodium falciparum* remains a common cause of mortality and morbidity (World Malaria Report 2022, 2022). All symptoms of the disease are caused by infection of

host erythrocytes, in which the parasite undergoes a well-defined 48-h developmental process. The first half of the intraerythrocytic cycle is commonly referred to as the 'ring' stage, as the parasites assume a ring-shaped morphology on Giemsa-stained smears. The parasite then matures to what is referred to as the 'trophozoite' form. At this stage the infected erythrocyte is less deformable and adheres to the endothelial cells lining the capillaries, thereby sequestering it from the circulation. The parasite subsequently initiates DNA replication and undergoes nuclear division during the final 'schizont' stage, culminating in the formation and release of the infectious progeny, the 'merozoites'. After their release from the host erythrocyte, in a process called egress (Dvorin and Goldberg, 2022; Tan and Blackman, 2021), merozoites bind to and invade erythrocytes to propagate the infection.

As erythrocytes infected with ring-stage parasites are present in the peripheral circulation, the parasites need to adapt to an important host defence against blood-borne pathogens: removal of infected erythrocytes in the spleen. In the red pulp of this organ, blood is filtered through extremely narrow passages (~0.65 μm) called interendothelial slits (IES). Passage through the IES requires that erythrocytes are highly deformable. Erythrocytes with decreased deformability, such as senescent and phenotypically altered erythrocytes, are removed (Mebius and Kraal, 2005; Thiagarajan et al, 2021; Cranston et al, 1984). The importance of the spleen for the control of malaria infection is evident in asplenic individuals. Whereas parasites present in the circulation of individuals with a spleen are mostly ring-stage parasites, trophozoites and schizonts are readily observed in the circulation of splenectomized individuals (Garnham, 1970). However, even erythrocytes containing ring-stage parasites are susceptible to removal by the spleen (Safeukui et al, 2008), with the most 'spherocytic' (spherical) erythrocytes preferentially filtered out (Safeukui et al, 2013).

Although the development of malaria parasites inside the host erythrocyte has been studied for well over a century, the events leading to the establishment of the parasite inside the host cell have not been investigated in detail, owing to the difficulty in synchronizing the invasion events and, until recently, in genetic manipulation of

[1]Faculty of Infectious and Tropical Diseases, London School of Hygiene & Tropical Medicine, London WC1 7HT, UK. [2]Department of Biological and Medical Sciences, Oxford Brookes University, Gipsy Lane, Oxford OX3 0BP, UK. [3]INSERM-U1134, BIGR, Université Paris Cité and Université des Antilles, Paris, France. [4]Electron Microscopy Science Technology Platform, The Francis Crick Institute, London NW1 1AT, UK. [5]University College London, Institute of Ophthalmology, 15-43 Bath Street, London EC1V 9EL, UK. [6]Department of Hepatobiliary Surgery and Liver Transplantation, Hôpital Beaujon, AP-HP, Clichy, France. [7]Malaria Biochemistry Laboratory, The Francis Crick Institute, London NW1 1AT, UK. [8]School of Life Sciences, Keele University, Staffordshire ST5 5BG, UK. [9]Present address: Centre for Ultrastructural Imaging, King's College London, New Hunt's House, Guy's Campus, London SE1 1UL, UK. ✉E-mail: C.van.Ooij@keele.ac.uk

the parasites. Hence, how the parasite transforms from the extracellular form to the intracellular 'ring' form is not well understood. Even though this stage is referred to as a 'ring', it is widely appreciated that *P. falciparum* parasites can assume an amoeboid (dendritic) shape during the first part of the intraerythrocytic cycle (Liffner et al, 2023; Gilson and Crabb, 2009; Grüring et al, 2011; Bannister et al, 2004; Sakaguchi et al, 2016; Riglar et al, 2013). However, only two studies have investigated this form in some detail, which revealed that parasites can assume an amoeboid shape early during the intraerythrocytic cycle and freely interconvert between amoeboid and spherical forms (Grüring et al, 2011; Riglar et al, 2013). The amoeboid forms can have multiple, motile 'limbs' (Grüring et al, 2011; Gilson and Crabb, 2009; Sakaguchi et al, 2016; Bannister et al, 2004), but neither the mechanism and timing of formation of the amoeboid shape after invasion nor its role in the survival of the parasite in the host have been established. Inhibiting amoeboid formation with jasplakinolide, which affects actin dynamics, did not affect the replication of the parasite (Grüring et al, 2011), indicating that in vitro, transition to this shape is not required. However, the effect of jasplakinolide does indicate that the transition to the amoeboid form is an active process. In this study, we investigated the mechanism of the transition from the round shape of the merozoite to the amoeboid shape of the parasite after invasion, the role of the amoeboid shape in parasite growth and the involvement of PV6, a parasite phospholipid transport protein, a mutant of which has previously been shown to form aberrant rings in Giemsa-stained smears (Hill et al, 2016; Fréville et al, 2024; Dans et al, 2024). Combining volume (3D) electron microscopy (serial block-face scanning electron microscopy (SBF-SEM)), tight synchronization of parasite erythrocytes invasion and conditional mutagenesis, we observed that upon invasion, rather than passively transitioning to a 'ring' shape, *P. falciparum* parasites undergo a rapid and complex metamorphosis from the round merozoite to an amoeboid shape.

This process involves a parasite-driven expansion of its surface area that occurs in at least two distinct steps. Importantly, the parasite phospholipid transfer protein PV6 (PFA0210c/PF3D7_0104200, also referred to as PfSTART1 (Nguyen et al, 2024; Dans et al, 2024)), is essential to progress to the amoeboid form, affecting the second growth step. PV6-dependent modifications of the erythrocyte are important for the survival of the parasite in the host, as erythrocytes infected with mutants lacking this protein are less able to traverse the spleen. Our work describes in detail and provides mechanistic insight into the transformation of malaria parasites from merozoites to the amoeboid shape and reveals that the changes in the erythrocyte are important for survival in the host.

# Results

## Merozoites rapidly transform to an amoeboid shape after invasion

To establish when after invasion parasites transform into the amoeboid form, we observed parasites fixed at specific times after invasion. In these samples, parasites were detected in a variety of shapes (Fig. 1A). As parasites interconvert between amoeboid and round forms (Grüring et al, 2011), these different shapes likely represent intermediates of the interconversion process. Note that in this study, all non-round parasites, including those with a single

limb (flask-shaped) and square parasites, were scored as amoeboid (Fig. 1A). To ensure that invasion was highly synchronous, we used reversible egress inhibitors (Compound 2 and ML10) that allow egress (and hence invasion) of the parasites to be timed to within a very narrow time window by removal of the inhibitor (Collins et al, 2013b; Ressurreição et al, 2022, 2020; Baker et al, 2017); arrested parasites egress and invade approximately 15 min after removal of the inhibitor (Ressurreição et al, 2020) (Fig. EV1A). Amoeboid forms could be detected as early as 20 min after removal of the egress inhibitor, suggesting that parasites can assume an amoeboid shape within 5 min after invasion (Fig. 1B). Live-cell imaging confirmed this observation: when parasite invasion was viewed live, the average time from invasion to amoeboid formation was approximately 17.5 min (Fig. 1C,D; see Fig. EV1B for an additional example of an invading parasite), with several parasites transitioning to the amoeboid form within 10 min. The slight difference in the apparent rate of conversion to the amoeboid shape in the two experiments may reflect that the observation of the live parasites may have negatively affected the transformation. Hence, the conversion from merozoite to the amoeboid shape after invasion is very rapid, most often occurring within the first 20 min after invasion.

## Merozoites contain all the parasite components for the transformation to an amoeboid shape

To gain insight into the mechanism underlying transformation to the amoeboid shape, wild-type parasites were allowed to invade erythrocytes in the presence of either Hanks' Balanced Salt Solution (HBSS), which contains only salts and glucose, or culture medium supplemented with the protein synthesis inhibitor cycloheximide. When evaluated by live microscopy 2 h after removal of egress inhibitor, neither treatment had significantly affected the transformation to amoeboid shapes, with ~65% of intraerythrocytic parasites assuming an amoeboid shape under each condition (Fig. 1E,F), indicating that neither external nutrients nor de novo protein synthesis are required for the transformation of parasites into amoeboid shapes following invasion. Hence, merozoites appear to contain all the necessary parasite components to drive this transformation.

To determine whether other *Plasmodium* species form amoeboid shapes, we observed *Plasmodium knowlesi* parasites 1–3 h after invasion. Amoeboid shapes were detected, albeit at a much lower frequency compared to *P. falciparum* (~10% vs ~65%) (Fig. 1F). Instead, most *P. knowlesi* parasites formed a cup shape (Fig. 1G), as has been reported previously (Liu et al, 2019). The *P. knowlesi* amoeboid parasites appeared to be as flexible and able to interconvert between amoeboid and spherical forms as *P. falciparum* parasites (Grüring et al, 2011) (Movies EV1, EV2).

## Parasites lacking PV6 cannot assume an amoeboid shape

We and others have previously shown that parasites lacking PV6 or parasites in which PV6 is inhibited remain small after invasion compared to wild-type and untreated parasites (Hill et al, 2016; Fréville et al, 2024; Dans et al, 2024). However, previous studies did not examine whether this reflects a collapse of the parasite or whether a parasite lacking PV6 activity simply fails to grow. To address this, we used a rapamycin-inducible *pv6* mutant in which

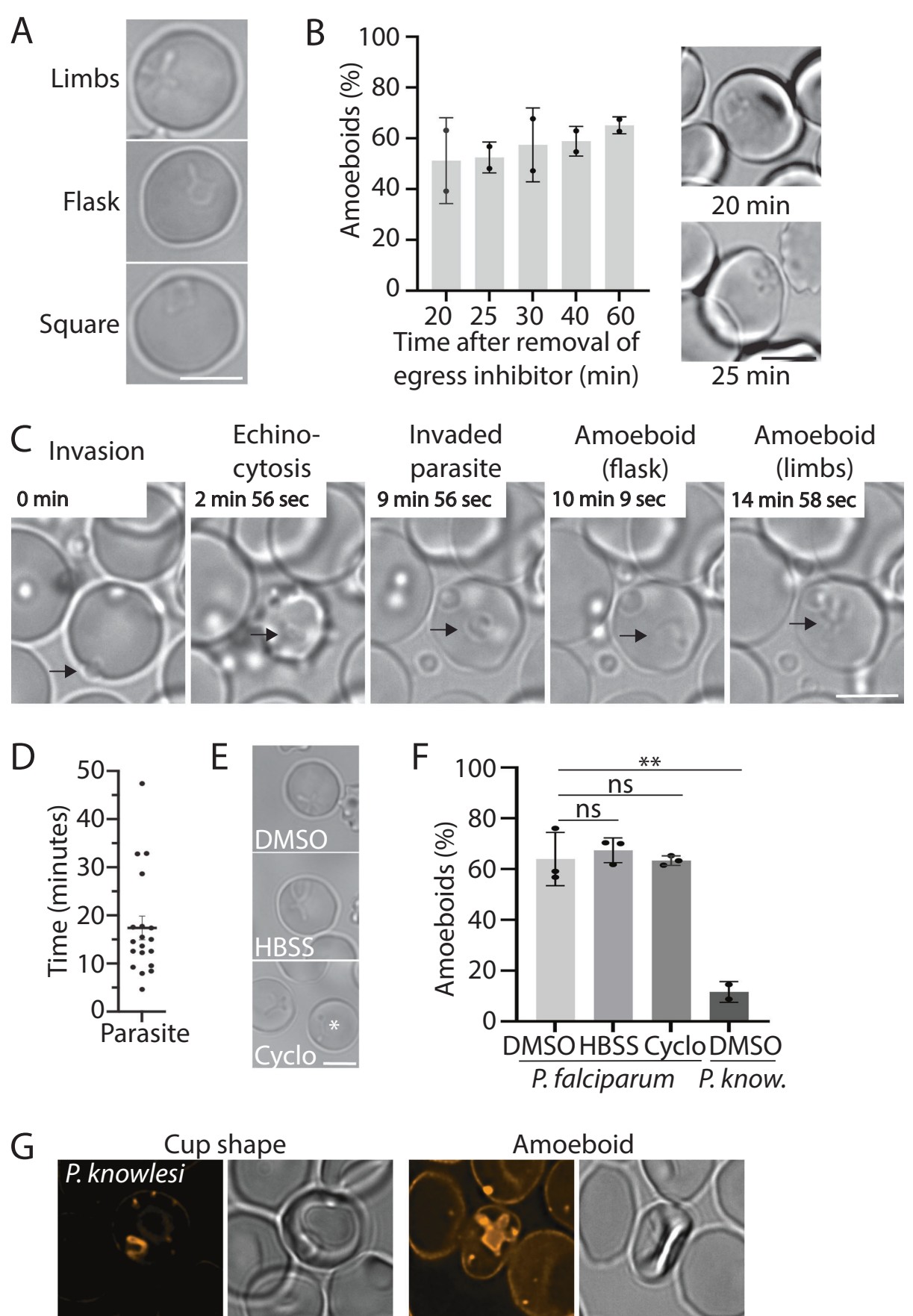

**Figure 1. Amoeboid formation of *Plasmodium* parasites during the early erythrocytic cycle.**

(A) Morphologies of *P. falciparum* parasites that were scored as amoeboid forms. (B) Timing of amoeboid formation after invasion. Parasites were fixed at the indicated time after removal of ML10 and amoeboid formation was subsequently scored by microscopy. Results shown are the combination of two biological replicates; error bars: $+/-$SD. Panels on the right are examples of amoeboid forms detected at the indicated time. (C) Live-cell imaging of amoeboid formation. The frame rate was one frame/second, time starting from invasion is indicated in each panel. The arrows indicate the parasite. (D) Timing of amoeboid formation in 19 parasites observed from the moment of invasion by video microscopy; error bars: $+/-$SEM. (E, F) Amoeboid formation of *Plasmodium falciparum* parasites after invasion in the presence of either DMSO in cRPMI (DMSO), Hanks' Balanced Salt Solution (HBSS) or 5 μM cycloheximide (Cyclo) in cRPMI 2 h after removal of ML10 and *Plasmodium knowlesi* 1–3 h after removal of ML10; error bars: $+/-$SD. (E) Representative images of live *Plasmodium falciparum* parasites. The (*) indicates an oddly shaped elongated parasite that was occasionally detected in the cycloheximide-treated samples. (F) Amoeboid formation in the indicated condition. Data represent three biological replicates of a minimum of 100 parasites (*P. falciparum*) and two biological replicates of *Plasmodium knowlesi* of at least 50 parasites (two-tailed t-test: ns: not significant; **P < 0.01 (P = 0.0076), error bars: $+/-$SD. (G) Live-cell fluorescence imaging of *Plasmodium knowlesi*-infected erythrocytes stained with Bodipy C5-ceramide 2 h after removal of egress inhibitor. All scale bars represent 5 μm. Source data are available online for this figure.

*pv6* is excised after addition of rapamycin, allowing the phenotype of parasites lacking PV6 to be analysed, even though PV6 is essential for parasite proliferation (Hill et al, 2016). When trophozoites were treated with rapamycin approximately 30 h after invasion, they matured seemingly normal prior to egress—they produced the same number of merozoites, had the same DNA content at the schizont stage and survived similarly to DMSO-treated parasites and the resulting merozoites invaded at the same rate (Appendix Fig. S1A–D). In contrast, the development of the parasites after invasion was severely affected. Measurement of the area of recently invaded parasites in Giemsa-stained smears revealed no difference in the size of wild-type (DMSO-treated) and mutant (rapamycin-treated) parasites 20 min after removal of the egress inhibitor, at which point the parasites had invaded the host cell approximately 5–10 min prior (Ressurreição et al, 2020). Thirty minutes after removal of the egress inhibitor, mutant parasites were significantly smaller than the wild type and did not form standard ring forms when detected in Giemsa-stained smears (Fig. 2A,B; Appendix Fig. S1E, Appendix Table S1). The size of the parasites lacking PV6 did not increase further, indicating that these parasites do not develop further after establishing themselves in the host erythrocyte. Hence, PV6 most likely functions immediately after invasion. Despite their small size, parasites lacking PV6 remained viable for an extended period after invasion, as indicated by positive MitoTracker staining (Appendix Fig. S1D). Similar survival of parasites was observed after treatment with a PV6 (PfSTART1) inhibitor (Dans et al, 2024). This makes PV6 one of the first proteins known to be required immediately following invasion of the host cell, along with the inner membrane complex protein IMC1g and an unidentified Plasmepsin V target protein, both of which also affect parasite development at this stage (Cepeda Diaz et al, 2023; Fréville et al, 2024; Polino et al, 2020).

Importantly, observation of wild-type and mutant parasites by live microscopy showed that wild-type parasites assumed an amoeboid shape after invasion but that parasites lacking PV6 did not—these parasites were smaller and more spherical (Fig. 2C–E). Although the mutant parasites did not move as much as the wild-type parasites inside the erythrocyte, these parasites could be detected moving slightly. However, they seemingly remained at the periphery of the erythrocyte, leaving open the possibility that despite the movement of the parasite, the PVM remains attached to the erythrocyte plasma membrane (Movies EV3, EV4). Although the wild-type parasites were motile, with limbs that moved relative to the rest of the parasite, the positioning within the erythrocytes did not change appreciably (Movies EV5, EV6). As treatment of wild-type parasites with jasplakinolide (which prevents amoeboid formation (Grüring et al, 2011)) affects the shape, but not the area, of wild-type parasites (Fig. 2C–E), we conclude that the smaller size of the mutant parasites does not reflect an inability to transform into the amoeboid shape but rather an inability to increase in size. Hence, PV6 is likely required to increase the surface area of the parasites that then allows the parasite to assume an amoeboid shape.

## The transition from merozoite to the amoeboid shape is a two-step process promoted by a rapid increase in surface area

To investigate the changes in parasite shape in detail, we applied SBF-SEM—a 3D volume electron microscopy technique—to obtain series of hundreds of consecutive EMs sections separated by 50 or 70 nm (Movies EV7–EV9). These data were then used to generate 3D models of parasites and erythrocytes. We focused on two time points: 20 min and 2 h after the removal of egress inhibitor and compared wild-type parasites with parasites lacking PV6 at these times. Flattened wild-type parasites resembling the amoeboid form could already be detected 20 min after removal of the egress inhibitor, although only few parasites had formed multiple limbs at this time point (Fig. 3A, see Fig. EV2 for additional views and EV3 for additional models). Two hours after removal of the egress inhibitor wild-type parasites had assumed various shapes, some with multiple limbs, consistent with the finding that these parasites are very motile and actively interconvert between amoeboid and spherical forms (Grüring et al, 2011). Interestingly, all limbs of the amoeboid parasites appeared to be attached to a central section of the parasite and no further bifurcation of the limbs was detected in any parasite, which potentially indicates that the parasite contains one central organizing centre from which the limbs extend.

In contrast to wild-type parasites, parasites lacking PV6 did not possess limbs or assume an amoeboid shape. These parasites were more spherical than wild-type parasites but did assume somewhat extended forms 20 min after removal of egress inhibitor. However, 2 h after egress inhibitor removal, these parasites had become nearly completely spherical (Fig. 3A; Fig. EV2 for additional views and Fig. EV3A for additional models, Movies EV7–EV9), indicating that PV6 is required for the progression to the amoeboid shape.

The presence of late-stage schizonts containing fully formed, segregated merozoites in the samples obtained 2 h after removal of the egress inhibitor allowed individual merozoites to be modelled as well (Fig. 3B). Furthermore, the infected erythrocyte could be

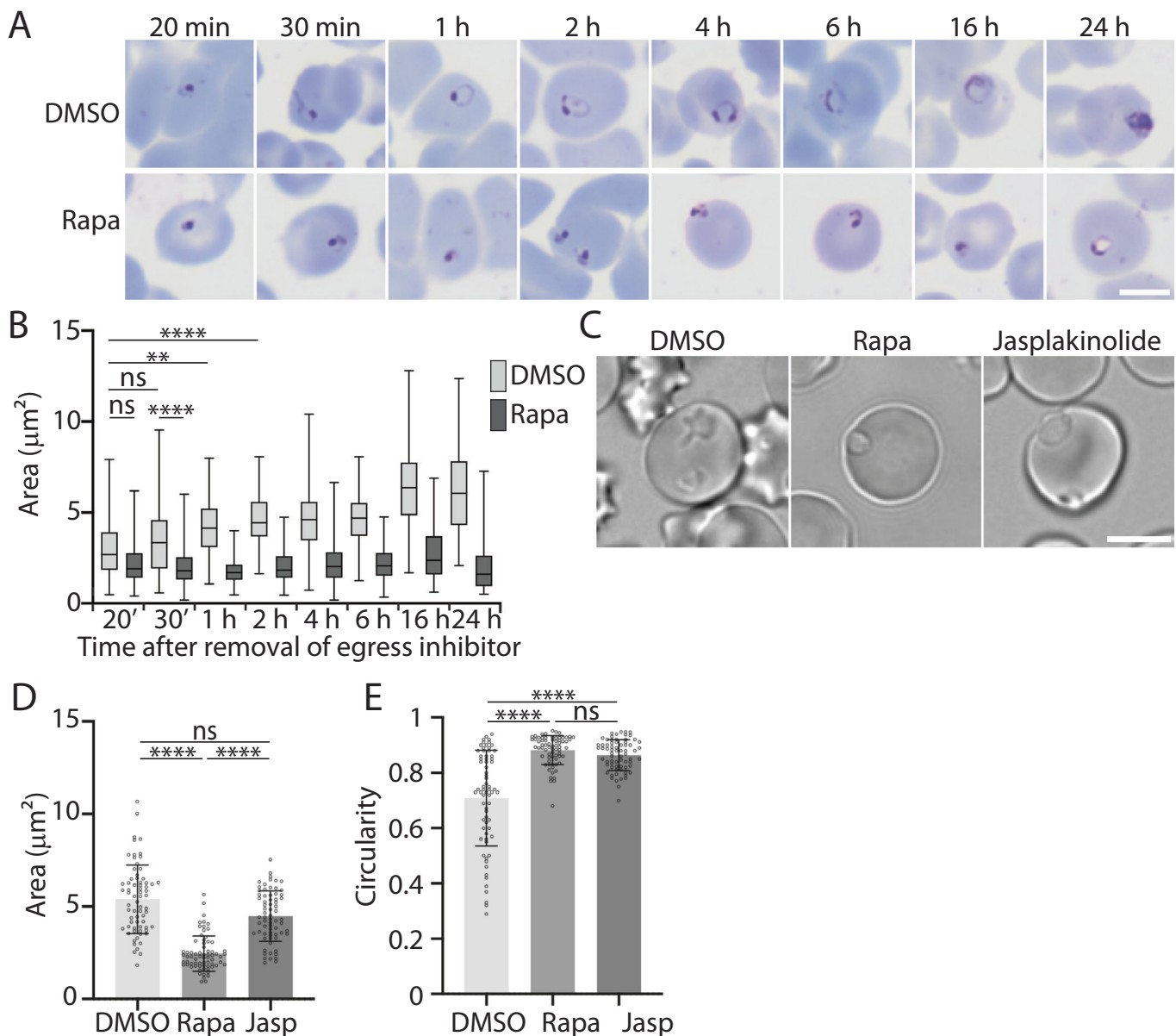

**Figure 2. Development of wild-type parasites and parasites lacking PV6 at early time points post-invasion.**

(A) Giemsa-stained smears of PV6 DiCre (PfBLD529) parasites treated with DMSO or rapamycin (Rapa). Development of synchronized parasites was examined in the cycle following DMSO or rapamycin treatment at the indicated time after removal of egress inhibitor (see EV1 for outline of experiment). Scale bar represents 5 μm. (B) Size (area) of DMSO and rapamycin-treated PV6-diCre parasites. Data represent three biological replicates with a minimum of 20 parasites per sample. Box represents the interquartile range and the horizontal line within the box represents the median. Whiskers indicate range from minimum to maximum. Significance was determined using Kruskal–Wallis test: ns: not significant; *P < 0.05, **P < 0.01, ***P < 0.001, ****P < 0.0001. For additional statistical analyses and exact p values, please see Appendix Table S1. (C) Live-cell imaging of PV6-DiCre (PfBLD529) parasites treated with DMSO or rapamycin in the previous cycle or with jasplakinolide for 1 h prior to imaging. Parasites were imaged 2 h after removal of ML10. Scale bar represents 5 μm. (D) Quantification of the size (area) of the parasites in the experiment presented in (C). Data are based on measurement of at least 20 rings in each of the three biological replicates performed. Rapa: rapamycin-treated; Jasp: jasplakinolide-treated (Kruskal–Wallis test: ns: not significant; ****P < 0.0001; ns; not significant; error bars: +/−SD). (E) Analysis of the circularity of the parasites in the experiment presented in (C). Rapa: rapamycin-treated; Jasp: jasplakinolide-treated (Kruskal–Wallis test: ns: not significant; ****P < 0.0001; error bars: +/−SD). Source data are available online for this figure.

modelled, revealing that the parasite occupies most of the erythrocyte (Figs. 3B and EV2).

The 3D models of the invaded parasites and the merozoites allow the surface area of the parasites to be determined and the growth of the parasite to be quantitated. This revealed that the

surface area of wild-type parasites increased between the 20-min and the 2-h time points, consistent with the analysis of Giemsa-stained parasites (Fig. 3C). At 20 min after removal of the egress inhibitor, parasites lacking PV6 have the same surface area as wild-type parasites, but their surface area failed to increase by the 2-h

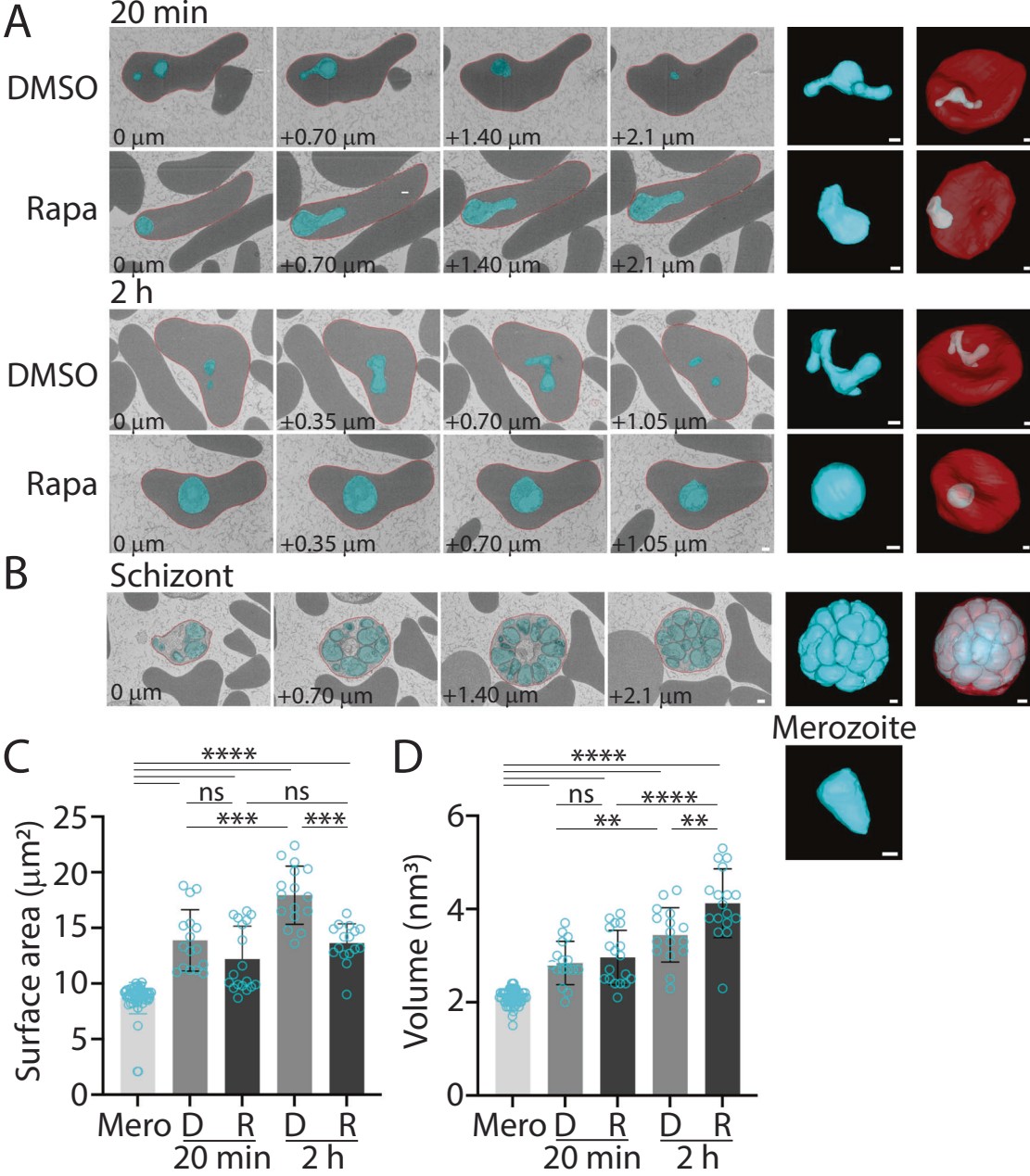

**Figure 3. Amoeboid formation is accompanied by a large increase in surface area and requires PV6.**

(A) SBF-SEM slices (sections) and surface renderings of PV6-DiCre (PfBLD529) parasites treated with DMSO and rapamycin 20 min and 2 h after removal of egress inhibitor, illustrating the shape and positioning of the parasite (cyan) within the erythrocyte (red). Indicated in the panels is the distance between the sections. The panel second from right shows a 3D model of the parasite shown in the sections on the left-hand side. On the far right is shown the parasite modelled inside the erythrocyte. Scale bar represents 500 nm. (B) Modelling of schizont and individual merozoite. (C) Surface area of merozoites (Mero) and DMSO-treated (D) and rapamycin-treated (R) parasites at the indicated time after removal of egress inhibitor. Data represent the measurement of at least 15 parasites. The Mann–Whitney U test was performed for statistical analysis: ns-not significant; ***$P < 0.001$ ($P = 0.0004$ for comparison DMSO 20 min-DMSO 2 h and $P = 0.0001$ for comparison DMSO 2h-RAPA 2 h); ****$P < 0.0001$, Error bars $+/-$ SD. See Fig. EV2 for additional views of these parasites and Fig. EV3 for additional models. (D) Volume of merozoites (Mero) and DMSO-treated (D) and rapamycin-treated (R) parasites at the indicated time after removal of egress inhibitor. Data represent the measurement of at least 15 parasites. The Mann–Whitney U test was performed for statistical analysis: ns-not significant; **$P < 0.01$ (($P = 0.0034$ for comparison DMSO 20 min-DMSO 2 h and $P = 0.0051$ for comparison DMSO 2h-RAPA 2 h); error bars $+/-$ SD. See Fig. EV2 for additional views of these parasites and Fig. EV3 for additional models. Source data are available online for this figure.

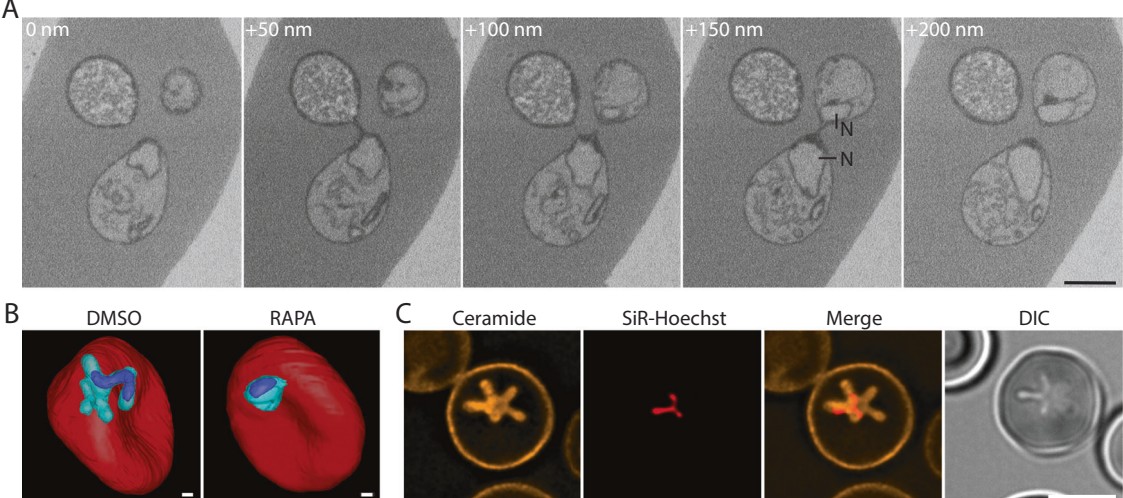

**Figure 4. Connections between limbs of the amoeboid-shaped parasites are very narrow and can separate the nucleus into lobes.**

(A) Consecutive SBF-SEM sections, showing three limbs of a parasite and the thin connections between them. Numbers in upper left-hand corner of the panels indicate distance between the sections, N indicates the nucleus and the scale bar represents 500 nm. (B) Three-dimensional models of infected erythrocytes (red) 2 h after removal of egress inhibitor, showing the nucleus (dark blue) in a wild-type (DMSO) parasite and a parasite lacking PV6 (RAPA); parasites are shown in cyan. Scale bar represents 500 nm. (C) Live-cell imaging of an infected erythrocyte stained with ceramide and the DNA dye SiR-Hoechst. Infected erythrocytes were imaged 2 h after the removal of egress inhibitor to reveal the ceramide and DNA staining. Also shown is the cell imaged using DIC. The imperfect alignment of the ceramide and SiR-Hoechst staining is the result of the movement of the parasite during imaging. Scale bar represents 5 μm. Source data are available online for this figure.

time point (Fig. 3C). The volume of wild-type and mutant parasites also increased over time; interestingly, the volume of mutant parasites increased more than that of wild-type parasites (Fig. 3D).

Similar results were obtained in a separate SBF-SEM experiment that used parasites that had been synchronized at the start of the first infection cycle but were not synchronized before entry into the second infection cycle (Fig. EV1). In this experiment, wild-type parasites also assumed amoeboid shapes, whereas parasites lacking PV6 were spherical (Fig. EV3B) and had a smaller surface area than wild-type parasites (Fig. EV3C), however, in this experiment, no volume increase was observed (Fig. EV3D). This discrepancy may be due to the lower level of synchronicity of the parasites used in that experiment. Nevertheless, these results show that the formation of the amoeboid form requires a PV6-mediated increase of the parasite surface area.

Interestingly, the surface area of merozoites was significantly smaller than that of intraerythocytic parasites, both wild type and lacking PV6, observed 20 min after removal of the egress inhibitor. This indicates that the parasites undergo a rapid expansion between release from the schizont and several minutes after the completion of invasion (Fig. 3C). The transformation of merozoites to the amoeboid shape therefore consists of at least two steps: a transformation of a merozoite to a recently invaded parasite, followed by an expansion of the surface area and transformation to the amoeboid shape. This second step does not occur in the absence of PV6, indicating that this protein has an essential role in the progression of the development of the parasites to this stage.

## Parasites with an amoeboid shape have an elongated nucleus

Close analysis of the individual sections of the SBF–SEM experiments revealed that the connection between the limbs of the parasite can be extraordinarily thin. In the consecutive SBF-SEM sections shown in Fig. 4A, three limbs of the parasite are visible, with a connection detected between the central limb and the other two limbs in only two 50 nm sections, with the connection barely perceptible in one of those two sections. In this parasite, the connections between the limbs is thus likely less than 100 nm wide. Interestingly, the nucleus in this parasite spanned one of these very narrow connections, suggesting that the nucleus is flexible and can span multiple limbs. Although uncommon, such narrow connections between limbs were not unusual, although not all of them contained the nucleus (see additional examples in Fig. EV4A).

Modelling of the nucleus in several SBF-SEM parasite models confirmed that the nucleus can extend into multiple limbs and also form an elongated, curved structure that fills one limb (Figs. 4B and EV4B), as also seen in previous 3D reconstructions (Bannister et al, 2004; Sakaguchi et al, 2016). Fluorescence microscopy of live parasites stained with a DNA dye 2 h after the removal of egress inhibitor similarly revealed elongated nuclei that protrude into multiple limbs (Figs. 4C and EV4C). Even in rounder parasites, the nucleus was extended or appeared to be circular, rather than spherical (Fig. EV4C, right-hand panels). This change in nuclear morphology likely starts after invasion has been completed, as the nucleus of invading merozoites has been found to form a barrier during invasion, slowing down the invasion process when it passes the tight junction (Rosario et al, 2019). Supporting this, in some amoeboid-shaped parasites, the nucleus was still spherical and formed the thickest region of the parasite (Fig. EV4D, also see the DMSO-treated parasite in Fig. 4B), suggesting that formation of the amoeboid form can occur prior to nuclear remodelling. However, the nuclei in parasites lacking PV6 were more spherical than the nuclei in wild-type parasites, indicating that amoeboid formation

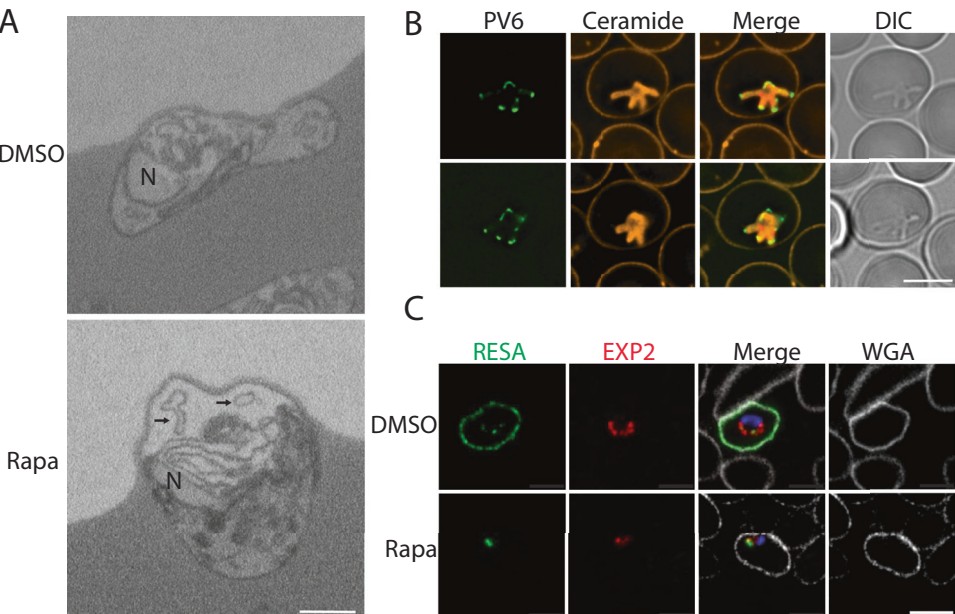

**Figure 5. Phenotype of *P. falciparum* parasites lacking PV6.**

(A) SBF-SEM sections from DMSO-treated (upper panel) and rapamycin-treated (lower panel) PV6 DiCre parasite culture, showing the accumulation of membrane whorls around parasites lacking PV6. Arrows indicate some of the lipid whorls. See Fig. EV4A for additional images. (B) Live-cell fluorescence imaging of parasites expressing an mNeonGreen-HA₃-PV6 fusion (PfBLD717) one hour after removal of ML10. Membranes were labelled using Bodipy C5-ceramide (orange). (C) Immunofluorescence staining of DMSO-treated and rapamycin-treated PV6-DiCre parasites 6 h after removal of ML10 using anti-RESA (green) and anti-EXP2 (red) antibodies. DNA was stained with Hoechst 33342 (blue) and the erythrocyte plasma membrane was visualized with Wheat Germ Agglutinin (WGA)-Alexa Fluor 647 (white). In all panels, the scale bar represents 5 μm. Source data are available online for this figure.

may be required for nuclear remodelling (EV4A). Similar changes to nuclear morphology have been detected in neutrophils, which undergo extreme shape changes as they cross the endothelium (Salvermoser et al, 2018). As is the case in those cells, the change in the nuclear morphology in parasites likely promotes the deform-ability of the parasite.

## Membrane 'whorls' accumulate around parasites lacking PV6

In SBF-SEM sections, parasites lacking PV6 were often detected close to the edge of the infected erythrocyte and in many cases whorls of what appeared to be lipid (membrane) material were detected in the space between the parasite and the parasitophorous vacuole membrane (PVM) (Fig. 5A; see Fig. EV5A for additional images). In erythrocytes infected with wild-type parasites, whorls were detected in some cases, although the frequency was lower (18% vs 60% in erythrocytes infected with parasites lacking PV6) (Table 1) and the size of the area of the whorls appeared smaller, although this was difficult to quantitate. As PV6 is a phospholipid transfer protein, the smaller surface area of the mutant parasites (and hence the PVM) and the accumulation of membrane whorls in the parasitophorous vacuole are consistent with the hypothesis that PV6 has a role in the transfer of lipids to the PVM to support the rapid expansion of parasites (Hill et al, 2016; van Ooij et al, 2013; Dans et al, 2024). Possibly, wild-type parasites release lipids that are subsequently incorporated into the membranes by PV6. The whorls detected in wild-type parasites may thus represent a brief

intermediate stage before PV6 transfers the lipids from the whorls to other membranes.

## PV6 activity impacts the remodelling of erythrocytes following parasite invasion

To determine whether PV6 is in the correct subcellular location to distribute lipids to the PVM, the endogenous *pv6* gene was replaced with a form encoding a PV6-mNeonGreen-HA₃ fusion protein (Appendix Fig. S2). These modified parasites grew at a similar rate to wild-type parasites, indicating that the modification has no effect on the activity of the protein; a similar modification with an HA₃ tag at the same position in the protein had no effect on parasite growth (Fréville et al, 2024). The fusion protein was detected at the tips of the limbs of the amoeboid-shaped parasites, indicative of a localization in the PV, supporting previous findings using anti-PV6 antibodies (Fig. 5B) (Fréville et al, 2024).

Next, we aimed to determine whether parasites lacking PV6 modify the host cell similarly to wild-type parasites. Six hours after the removal of egress inhibitor, as expected, the exported protein RESA was readily detected at the periphery of erythrocytes infected with wild-type parasites (Foley et al, 1990), however, little to no export of RESA was detected in erythrocytes infected with parasites lacking PV6, indicating that little to no export occurs in the absence of PV6 (Figs. 5C and EV5B). The PVM protein EXP2 (Fischer et al, 1998), which mediates protein export (Mesén-Ramírez et al, 2016; Ho et al, 2018; de Koning-Ward et al, 2009), appeared to have a more restricted internal localisation in the PV containing parasites

**Table 1.** Detection of whorls in EM samples.

| Treatment | Parasites | Whorls | % (95% CI) |
|---|---|---|---|
| DMSO | 27 | 5 | 18.5 (6.30–38.08) |
| Rapamycin | 35 | 21 | 60.0 (42.11–76.13) |

*CI* confidence interval.

lacking PV6 (compared to a more peripheral localisation in the PV of wild-type parasites), indicating that EXP2 may not be in the correct location to export proteins (Fig. 5C). Interestingly, Riglar et al, showed that protein export does not appear to commence until about 10 min after invasion (Riglar et al, 2013). Hence, parasites need to undergo a particular development after invasion to become competent to export proteins and PV6 seems to be required to complete this development. Alternatively, PV6 may have a role in export itself, but no evidence for such a role has been published.

It has been shown previously that infected erythrocytes have a smaller surface area than uninfected erythrocytes, as some of the erythrocyte membrane is internalized to form the PVM (Geoghegan et al, 2021; Safeukui et al, 2013). Using imaging flow cytometry, we also observed this phenomenon (Fig. 6A; Appendix Table S2). This analysis also revealed that the surface area of erythrocytes infected with mutant parasites is ~4% smaller than erythrocytes infected with wild-type parasites when measured 30 min after the removal of egress inhibitor (Fig. 6B, Table 2, Appendix Fig. S3). This difference in surface area was slightly smaller, but still present, 90 min after the removal of egress inhibitor (Table 3). The difference between the DMSO-treated and rapamycin-treated infected erythrocytes was larger than the difference between the uninfected erythrocytes in the DMSO-treated and rapamycin-treated samples (Fig. 6B). Hence, PV6 activity mitigates the loss of surface area of infected erythrocytes. As smaller erythrocytes are more likely to be removed in the spleen (Pivkin et al, 2016; Safeukui et al, 2013), PV6 thus may have an important role in preventing the removal of infected erythrocytes in the spleen by decreasing the loss of surface area.

### Absence of PV6 decreases passage of infected erythrocytes through the spleen

To determine whether the phenotypic changes of parasites lacking PV6 and their host cells affect the ability of the parasite to escape clearance by the spleen, we applied erythrocytes infected with wild-type parasites and parasites lacking PV6 to a microsphiltration column, which mimics filtration of erythrocytes through the IES in vitro (Deplaine et al, 2011; Duez et al, 2015). Erythrocytes containing parasites lacking PV6 were retained in the column at a significantly higher rate than erythrocytes containing wild-type parasites (Fig. 6C), suggesting that erythrocytes containing mutant parasites are more likely to be removed from the circulation when crossing the spleen filtration barrier. Infected erythrocytes treated with jasplakinolide passed through the microsphiltration column similar to untreated parasites (Fig. EV6A), which may indicate that the decreased passage of the erythrocytes infected with parasites lacking PV6 is the result of the decreased surface area, rather than the inability to assume an amoeboid shape. However, it remains to be determined how flexible the jasplakinolide-treated parasites are.

When we further tested the ability of erythrocytes infected with either wild-type parasites or parasites lacking PV6 to pass through the spleen directly by perfusing erythrocytes through an ex vivo human spleen, we found that erythrocytes infected with mutant parasites were cleared faster than erythrocytes infected with wild-type parasites (Figs. 6D and EV6B–D). These results indicate that the modifications in the infected erythrocyte owing to the activity of PV6 play an important role in the survival of the parasite in the human host, and to our knowledge, this makes PV6 the first identified parasite protein whose activity has a positive effect on the passage of infected erythrocytes through the host's spleen.

## Discussion

Together, our results reveal that the establishment of the malaria parasite *Plasmodium falciparum* in the host erythrocyte is a much more complex process than previously thought. Rather than passively adopting a ring shape after invasion, starting soon after (and possibly even during) the invasion process, the parasite undergoes a drastic metamorphosis, changing from a small, compact cell to a large, extended and flexible cell. All components required for this transformation are present in the merozoite and the host cell, as the transformation also takes place in the absence of nutrients or protein synthesis. In the approximately 17 min required for the metamorphosis to occur, the cell is likely radically remodelled, such that the amoeboid-shaped parasite has little resemblance to the invading merozoite. Adopting an extended amoeboid form increases the surface area-to-volume ratio, which allows the parasite to be much more flexible than the spherical merozoite. Interestingly, although *P. knowlesi* parasites adopt an amoeboid shape at a much lower frequency, the cup shape that many of these parasites assume will similarly have a high surface area:volume ratio and therefore allows them to be flexible, similar to *P. falciparum*. This flexibility likely promotes passage of infected erythrocytes through the spleen and the microvasculature. Similarly, our data also reveal that the nucleus becomes extended and in some parasites is distributed over several limbs of the amoeboid, sometimes spanning connections between limbs less than 100 nm wide. This extended shape of the nucleus likely adds to the flexibility of the cell as a whole and is a distinct change from the merozoite, which appears to have a rigid nucleus (Rosario et al, 2019). As gene expression is in part dictated by the 3D organization of the genome (Ay et al, 2014; Bunnik et al, 2019), this distribution of the genome among the different lobes of the nucleus may impact the transcriptome of the parasite.

Although the mechanism of the merozoite-to-amoeboid metamorphosis remains unknown, it is of interest that this metamorphosis consists of at least two steps, starting with an expansion of the parasite between the merozoite stage and the recently invaded parasite, followed by a second expansion that likely involves the formation of the amoeboid shape and requires the parasite phospholipid transfer protein PV6. Our finding that parasites lacking this protein have a smaller surface area and PVM and are more spherical may indicate that this protein has a role in the expansion of the PVM. These findings build upon the results presented by Dans et al, who used lattice light-sheet microscopy to show that inhibiting PV6 activity also results in more spherical parasites (Dans et al, 2024). As this protein is

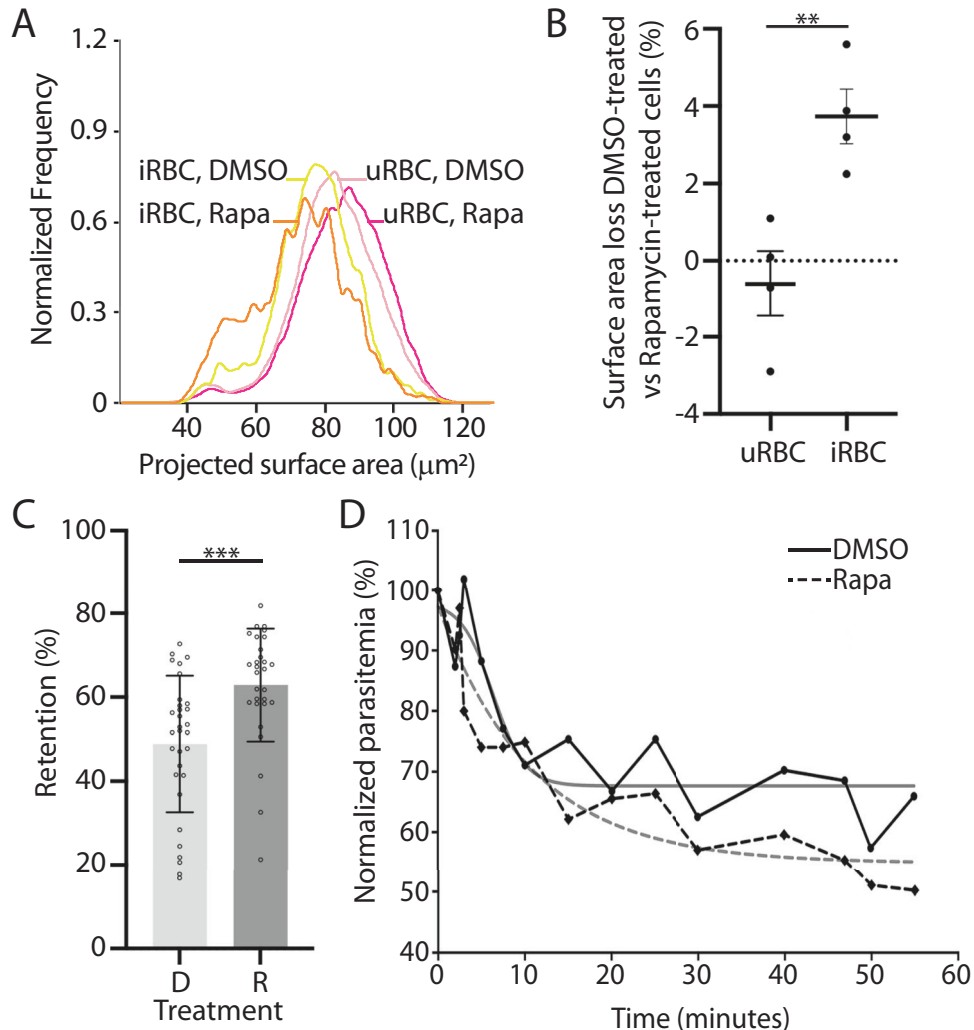

**Figure 6. Size and functional alteration of erythrocytes infected with parasites lacking PV6.**

(A) Surface area of uninfected erythrocytes (uRBCs) and erythrocytes infected with PV6-diCre parasites (iRBCs) 30 min after removal of egress inhibitor. The surface area of uRBCs and iRBCs in culture of wild-type (DMSO-treated) parasites and parasites lacking PV6 (Rapamycin-treated) was determined using imaging flow cytometry (Appendix Fig. S4). (B) Quantification of the surface area loss of uninfected (uRBC) and infected (iRBC) erythrocytes from cultures of DMSO-treated and rapamycin-treated parasites 30 min after removal of ML10, representing four experiments. Data represent four biological replicates (T-test; **$P < 0.01$ ($P = 0.0078$), error bars: $+/-$ SEM). (C) Retention of erythrocytes infected with wild-type (DMSO-treated – D) parasites and parasites lacking PV6 (Rapamycin-treated – R) on a microsphiltration column. Data represent 4 biological replicates with a minimum of 6 technical replicates. Significance was determined using Mann–Whitney test; ***$P < 0.0001$ ($P = 0.0001$), error bars: $+/-$SD. (D) Retention of erythrocytes infected with PV6-DiCre parasites treated with DMSO or rapamycin in an ex vivo human spleen. Blood containing erythrocytes infected with DMSO-treated (solid black line) and rapamycin-treated parasites (dashed black line) approximately 6 h after the removal of ML10 was continuously perfused through the spleen and the parasitemia was determined at the indicated times. Interpolation curves are shown in grey lines ($R^2 = 0.84$ for DMSO and $R^2 = 0.90$ for Rapa samples). Source data are available online for this figure.

present in the PV after invasion, it is well positioned to transfer parasite-derived phospholipids, either phospholipids from the parasite plasma membrane or phospholipids secreted from the rhoptries (Bannister et al, 1986; Stewart et al, 1986), to the PVM. This is consistent with a model in which the PVM is initially produced using lipids from the erythrocyte plasma membrane, and thus can form without PV6, and is subsequently expanded as the parasite increases in size during the transition to the amoeboid shape, using parasite-derived phospholipids (Geoghegan et al, 2021; Sherling and van Ooij, 2016). As PV6 is synthesized starting from approximately 20 h after invasion (Dans et al, 2024; van Ooij et al, 2013), it is possible that the

protein also functions during the later stages of the intraerythrocytic cycle. Treatment of infected erythrocytes with a PV6 inhibitor starting after the parasites have invaded did not seem to affect parasite development as investigated using Giemsa-stained thin films (Dans et al, 2024), possibly indicating that the protein does not function at this time, although further investigations may detect subtle defects that are difficult to detect using Giemsa staining. In addition, in one experiment our results indicated that although the surface area of parasites that lack PV6 is smaller than that of wild-type parasites, the volume of the parasites lacking PV6 was found to be larger. It is unclear what would cause this. The lack of export detected could be the

**Table 2.  Decrease in surface area of erythrocytes containing DMSO-treated or rapamycin-treated parasites 30 min after removal of egress inhibitor.**

| Sample | Strain | iRBC surface area loss (Rapa compared to DMSO) ($\mu m^2$) | iRBC surface area loss (Rapa compared to DMSO) (%) |
|---|---|---|---|
| 1 | PfBLD529-466 | 2.62 | 3.88 |
| 2 | PfBLD529 clone A8 | 1.54 | 2.25 |
| 3 | PfBLD529-466 | 4.34 | 5.6 |
| 4 | PfBLD529-466 | 2.36 | 3.2 |
| | Mean (Std Dev) | 2.7 (1.02) | 3.72 (1.22) |

result of a lack of EXP2 activity, which would affect the transport of solutes and water across the PVM.

The results presented are also consistent with a model in which phospholipids are transferred to the surface of the infected erythrocyte, as erythrocytes containing parasites lacking PV6 have a smaller surface area than erythrocytes infected with wild-type parasites. As erythrocyte size is an important determinant of passage through the spleen (Safeukui et al, 2013), the transfer of phospholipids to the erythrocyte surface likely increases the survival of the parasite in the host, as indicated by the increased clearance of erythrocytes containing parasites lacking PV6 in the spleen. However, the mechanism for this phospholipid transfer remains obscure, as there is no known connection between the PVM and the plasma membrane of the erythrocyte and PV6 is confined to the PV (Fréville et al, 2024). Potentially the decreased export of parasite proteins from parasites lacking PV6 may be the cause of the decreased mitigation of the loss of erythrocyte surface area. As round jasplakinolide-treated parasites can traverse a microsphiltration column with equal efficiency as untreated wild-type parasites, it is likely that the smaller surface area of the infected erythrocyte is the main cause of the increased loss of infected erythrocytes on a microsphiltration column and the ex vivo spleen.

The transformation of the merozoite to an amoeboid shape is a complex process that occurs in at least two distinct steps. PV6 is required for the parasite to undergo the second expansion and failure to progress leads to an inability of the parasite to export proteins. Previous results by Riglar et al, showed that export does not commence immediately after invasion, and instead, protein export was detected only approximately 10 min after invasion (Riglar et al, 2013). Potentially, the second step of the transformation to the amoeboid form is required for the PTEX export system to be activated or positioned in the PVM to allow for export.

Although the complexity of the metamorphosis of the parasite is readily apparent from the large changes in the shape and surface of the cell, this complexity is further underscored by the possible functions of the few proteins that have been identified to have a role in this process: a PV protein with phospholipid transfer activity (PV6), an inner membrane complex protein (IMC1g), an unidentified PM V target and a potential signalling protein (the phosphatase PP7) (Fréville et al, 2024; Cepeda Diaz et al, 2023; Patel et al, 2024; Polino et al, 2020). The metamorphosis from merozoite to amoeboid shape is an essential process, as parasites lacking PV6, IMC1g or PP7 do not develop, and in the case of

IMC1g and PP7, die rapidly during culture in vitro. Our finding that the metamorphosis is driven by parasite proteins already in the merozoite indicates that this process could represent a target for intervention strategies. As the host is more likely to clear erythrocytes that contain parasites lacking PV6, this protein may be a more promising drug target than previously thought (Dans et al, 2024).

Together, our results show that *Plasmodium falciparum* parasites undergo a rapid metamorphosis after invasion and that the alterations of the host cell that accompany this transformation increase the survival of the parasite inside the host.

# Methods

**Reagents and tools table**

| Reagent/Resource | Reference or Source | Identifier or Catalog Number |
|---|---|---|
| **Antibodies** | | |
| Anti Mouse IgG (H + L), Alexa Fluor 488, Invitrogen | Fisher Scientific | 10256302 |
| Anti Rabbit IgG (H + L), Alexa Fluor 568, Invitrogen | Fisher Scientific | 10463022 |
| Anti-RESA monoclonal antibody | Walter and Eliza Hall Institute Antibody Facility | |
| **Oligonucleotides and other sequence-based reagents** | | |
| PCR primers – see Methods for sequence | IDT DNA | |
| **Chemicals, Enzymes and other reagents** | | |
| In-Fusion HD Cloning Kit | Takara | 639648 |
| SYBR Green | Life Technologies | 10710004 |
| MitoTracker DeepRed | ThermoFisher | 12010156 |
| Hoechst 33342 | Bertin Pharma | 5547.25 MG |
| Bodipy C5-ceramide | Fisher Scientific | 11584167 |
| Wheat Germ agglutinin- Alexa Fluor 647 | Fisher Scientific | 11510826 |
| VectaShield | Vector | H-1000-10 |
| **Software** | | |
| 3dmod | https://bio3d.colorado.edu/imod/doc/3dmodguide.html | |
| FIJI | https://imagej.net/downloads | |
| GraphPad Prism | https://www.graphpad.com/features | |

## Parasite culture, parasite strains and rapamycin treatment

*P. falciparum* parasites were cultured in human erythrocytes (sourced from UK National Blood Transfusion Service and Cambridge Bioscience, UK) at 3% haematocrit maintained in RPMI-1640 (Life Technologies) supplemented with 2.3 g/L sodium bicarbonate, 4 g/L dextrose, 5.957 g/L HEPES, 50 $\mu$M hypoxanthine, 0.5% AlbuMax type II (Gibco), and 2 mM L-glutamine (cRPMI)

and incubated at 37 °C in the presence of 5% $CO_2$. *P. knowlesi* parasites were cultured in cRPMI further supplemented with 10% horse serum and was incubated in an atmosphere of 96% $N_2$, 1% $O_2$ and 3% $CO_2$ (Ressurreição et al, 2020).

For routine culture, *P. falciparum* parasites were synchronized by isolating late-stage parasites on a Percoll cushion (Rivadeneira et al, 1983), allowing the isolated parasites to invade fresh erythrocytes for 1–3 h and removing the remaining late-stage parasites on a second Percoll cushion. In the case of *P. falciparum*, any late-stage parasite remaining in the culture after the separation on the Percoll cushion were removed by treatment with 5% (w/v) sorbitol (Lambros and Vanderberg, 1979), leaving only recently invaded parasites. To allow invasion to take place in a shorter window, parasites were treated with 25 nM ML10 (from a 100 μM stock) or 1 μM Compound 2 (from a 1 mM stock) to prevent egress (Ressurreição et al, 2020; Collins et al, 2013b). When most schizonts appeared arrested, the parasites were pelleted, suspended in cRPMI containing erythrocytes at a haematocrit of 3% and incubated at 37 °C unless otherwise stated.

All experiments were performed with *P. falciparum* strains 3D7, PfBLD529 clone A8 (Hill et al, 2016) and PfBLD529-466. The two different PfBLD529 parasite lines contain the identical floxed *pv6* locus (*pfa0210c*/PF3D7_0104200) but differ in the location of the diCre cassette. In PfBLD529 clone A8 this is present at the *sera5* locus (Collins et al, 2013a) whereas in PfBLD529-466 it is inserted in the *pfs47* gene (Knuepfer et al, 2017). As previously described, the native *pv6* gene is excised upon treating PfBLD529 parasites with rapamycin (Hill et al, 2016).

Rapamycin treatment was performed on trophozoite-stage parasite approximately ~30 h post-invasion. Parasites were incubated with 10 nM rapamycin (from a 100 μM stock solution in DMSO) at 37 °C for 1 h and controls were treated with an identical volume of DMSO (Knuepfer et al, 2017; Collins et al, 2013a).

## Biosafety

The genetic manipulation of *Plasmodium* parasites took place at the Francis Crick Institute and the London School of Hygiene & Tropical Medicine (LSHTM). Approval for this work has been granted by the UK Health and Safety Executive (HSE) – project references in the Public Register are 542/07.1 (The Francis Crick Institute, *P. falciparum*), 654/01.2 (LSHTM, *P. falciparum*) and 654/15.1 (LSHTM, *P. knowlesi*).

## Parasite growth assay

To determine the parasite growth rate, the parasites were first tightly synchronized using a Percoll/Sorbitol strategy as described above. Thirty hours post-invasion, the cultures were treated with rapamycin or DMSO and adjusted to a parasitaemia of 0.1–0.2% and 2% haematocrit. Growth was assayed every two days for a total of 6 days by collecting a 50 μl aliquot which was fixed in an equal volume of fixative solution (8% paraformaldehyde, 0.2% glutaraldehyde in PBS) in a 96-well plate. When ready, the samples were prepared for analysis by flow cytometry: the fixative was removed and the cells were washed twice with PBS and suspended with 200 μl of PBS/SYBR Green 1 (1:5000, Life Technologies; Reagents Table) for 30 min in the dark. Finally, the samples were diluted in PBS (1:5) and the parasitaemia was measured using an Attune

cytometer (ThermoFisher) using the following laser settings: forward scatter 125 V, side scatter 350 V and blue laser (BL1) 530:30 280 V. Each experiment was performed in triplicate with a minimum of three biological replicates. Graphpad Prism v10 was used for statistical analysis.

## Quantification of merozoite number

The quantification of merozoites per schizont was conducted using Giemsa-stained thin smear of mature parasite enriched through a Percoll cushion (Rivadeneira et al, 1983). The parasites were imaged using an Olympus BX51 microscope equipped with an Olympus SC30 camera and a 100x oil objective, controlled by cellSens software. Each experiment was conducted in triplicate with a minimum of three biological replicates. Statistical analysis was performed using GraphPad Prism v10.

## Parasite invasion rate

Parasite invasion assays were performed as described previously (Mohring et al, 2020). Briefly, DMSO-treated and rapamycin-treated parasites were diluted to a parasitaemia of 1% and a haematocrit of 2%. At that point, a 50 μl starting sample (designated as H0) was collected and fixed in a PBS solution containing 8% paraformaldehyde, 0.01% glutaraldehyde. After a 24-h incubation period, a second sample was collected and fixed (designated as H24). The parasites were labelled with SYBR Green I (1:5000, Life Technologies; Reagents Table) and the parasitaemia was quantified using an Attune cytometer (Thermofisher; see detailed protocol above). Parasite invasion rate was determined as the ratio of H24/H0 parasitaemia. Each experiment was conducted in triplicate with a minimum of three biological replicates. Statistical analysis was performed using GraphPad Prism v10.

## DNA content analysis of schizonts

Highly synchronous schizonts were fixed and prepared as described above in the Parasite growth assay section. SYBR Green fluorescence was analysed using an Attune cytometer (ThermoFisher) using the following laser settings: forward scatter 125 V, side scatter 350 V and blue laser (BL1) 530:30 280 V. At least 100,000 cells were analysed per sample in three biological replicates. Data analysis was performed using FlowJo software.

## Live/dead staining

Parasites were treated with 10 nM rapamycin or an equivalent volume of DMSO for 30 min approximately 30 h after invasion. The cultures were adjusted to a parasitaemia of 5% and a haematocrit of 3%. Parasites were collected at that point, then the next day (Schizonts) and at 1, 4, 8, 24 and 32 h post-invasion. Upon collection, the samples were stained with 200 nM Mitotracker DeepRed (ThermoFisher; Reagents Table) for 15 min, washed with PBS and then fixed in an equal volume of fixative solution (8% paraformaldehyde, 0.2% glutaraldehyde in PBS) containing SYBR Green I (1:5000). After a minimum of 30 min, cells were washed with PBS and diluted in PBS (1:5) for cytometry. The analysis was performed on an Attune cytometer. Mitotracker-positive, SYBR Green-positive parasites were considered viable,

**Table 3.** Decrease in surface area of erythrocytes containing DMSO-treated or rapamycin-treated parasites 90 min after removal of egress inhibitor.

| Strain | Treatment | iRBC surface area loss compared to uRBC (%) | iRBC surface area loss (Rapa compared to DMSO) (%) |
|---|---|---|---|
| PfBLD529-466 | DMSO | 4.60% | |
| PfBLD529-466 | Rapamycin | 6.80% | 3.7% |
| PfBLD529 clone A8 | DMSO | 3.60% | |
| PfBLD529 clone A8 | Rapamycin | 5.80% | 2.8% |
| Mean | | | 3.26 |

Mitotracker-negative, SYBR Green-positive parasites were considered non-viable. The proportion of live parasite (stained with Mitotracker DeepRed) was compared to the proportion of all parasites (dead and viable) stained with SYBR Green as previously described (Amaratunga et al, 2014). Two biological replicates were performed, each performed in triplicate. Data analysis was performed using FlowJo software.

## Plasmids and transfection

Parasite line PfBLD529-466 was produced by transfecting 3D7 parasites with repair plasmid pBLD466 (also described as pBSPfs47DiCre) (Knuepfer et al, 2017). The resulting parasite line expresses diCre from the *pfs47* locus. This parasite line was subsequently transfected with plasmid pBLD529 (Hill et al, 2016) to produce PfBLD529-466. Parasite line PfBLD717 (producing the PV6-mNeonGreen fusion) was produced by amplifying the gene encoding mNeonGreen using primers CVO696b (CGCAGTCA ATCTCGAGTTAGTAAAGGAGAAGAAGATAATATGGCAAG) and CVO697b (CGTAAGGGTACTCGATTTATACAATTCATC CATTCCCATAACATCTGTAAATG). The resulting DNA fragment was cloned using InFusion (Takara Bio) into pBLD708 (Fréville et al, 2024) that had been digested with AvrII and SpeI, producing plasmid pBLD717. Transfections were performed using the schizont transfection protocol as described previously (Collins et al, 2013a). Briefly, schizonts purified on a Percoll cushion were allowed to invade fresh erythrocytes for 1–3 h. The residual schizonts were isolated on a Percoll cushion, pelleted, washed with cRPMI and suspended in a solution consisting of 100 μL AMAXA nucleofection P3 Primary Cell transfection reagent and 10 μL TE containing 15–30 μg targeting plasmid and 20 μg plasmid pDC2-Cas9-hDHFRyFCU (Knuepfer et al, 2017; Fréville et al, 2024) encoding Cas9 and the appropriate guide RNA. Transfection was carried out using the AMAXA nucleofection device (Lonza) using programme FP158. Immediately following transfection, schizonts were mixed with fresh erythrocytes and 3 ml cRPMI and incubated with shaking at 37 °C for 30 min to facilitate invasion. Drug selection with 2.5 nM WR99210 was started after 24 h and sustained for 7 days. Transfected parasites were typically recovered 3–4 weeks after transfection.

## Analysis of parasite development using Giemsa-stained samples

To analyse and measure the development of parasites, PfBLD529-466 (PV6 diCre) parasites were tightly synchronized using a Percoll-Sorbitol strategy and then treated with rapamycin and ML10 as outlined above. At the indicated time point after invasion, the parasites were pelleted, smeared on a microscope slide, fixed

with methanol and stained with Giemsa. The parasites were imaged using an Olympus BX51 microscope equipped with an Olympus SC30 camera and a 100x oil objective, controlled by CellSens software. Analysis of the images was performed using FIJI. The contour of each visible parasite was delineated using the FIJI freehand selection tool and the area was determined utilizing the area measurement function. Each experiment was performed in triplicate with a minimum of three biological replicates. Graphpad Prism v10 was used for statistical analysis.

Analysis of the ability of the parasite to form amoeboids was first performed using live microscopy. Parasites were tightly synchronized using the Percoll-reinvasion-Sorbitol strategy strategy detailed above. The parasites were treated with rapamycin at 30 h post-invasion and subsequently arrested at the late schizont stage with ML10 to inhibit egress (Ressurreição et al, 2020). When appropriate, the initiation of synchronized invasion was triggered by the removal of ML10. At the indicated time point after removal of ML10, the parasites were collected for analysis. Prior to imaging, the cells were labelled with 5 μg/mL Hoechst 33342 (Reagents Table) and allowed to settle in a 6-well Ibidi μ-slide with a glass bottom. Parasites were observed using a 100X oil immersion objective on a Nikon Ti-E inverted microscope equipped with a Hamamatsu ORCA-Flash 4.0 Camera and Piezo stage driven by NIS elements version 5.3 software. Image J (Reagents Table) was employed for image processing, with adjustments made to brightness and contrast using the reset command for autoscale. Images were sized in Photoshop and figures were produced using Illustrator. To measure the area and circularity of the parasites, the contour of each visible parasite was delineated using the FIJI freehand selection tool followed by additional measurements using the area and circularity functions. Circularity values range from 0 to 1 (derived from the area and perimeter of the cell). Higher values indicate shapes of increasing circularity, with a perfect circle characterised by a value of 1. Each experiment was performed in triplicate with a minimum of three biological replicates. Graphpad Prism v10 was used for statistical analysis.

To visualize parasites using fluorescent ceramide, the culture was incubated at 37 °C in the presence of 7 mM Bodipy C5-ceramide in complex with BSA (diluted from a 100 mM stock in PBS) (Fisher Scientific; Reagents Table) and 2 μM 5-SiR-Hoechst (Lukinavičius et al, 2015) in RPMI without Albumax for a minimum of 20 min. The culture was pelleted at $2000 \times g$, suspended in RPMI without Albumax and diluted to a haematocrit of 0.6% using RPMI without Albumax before loading into a poly-L-lysine-coated 6-well Ibidi μ-slide. The parasites were visualized on a Nikon Ti-E inverted microscope as described above.

To determine the effect of the protein synthesis inhibitor cycloheximide or Hanks' Balanced Salt Solution containing $Mg^{2+}$ and $Ca^{2+}$ (HBSS) on amoeboid formation, synchronized late-stage

parasites were purified on a Percoll cushion, washed with, and then suspended in, cRPMI and incubated in the presence of ML10 to prevent egress. When most parasites appeared arrested, 5 μM cycloheximide or an equivalent volume of DMSO was added and the parasites were incubated for an additional hour. The parasites were then pelleted, mixed with erythrocytes, in the presence of either DMSO or cycloheximide. Half of the DMSO-treated parasites was removed, pelleted, washed four times with HBSS and suspended in HBSS and all samples were incubated at 37 °C to allow invasion to take place. After two hours, the parasites were diluted to a haematocrit of 0.6% in RPMI before being loaded into a poly-L-lysine-coated 6-well Ibidi μ-slide. The parasites were visualized live on a Nikon Ti-E inverted microscope as described above.

To determine the timing of amoeboid formation, late-stage parasites were purified on a Percoll cushion and incubated at 37 °C in the presence of 10 nM ML10 to prevent egress. When most parasites appeared arrested, the parasites were pelleted and mixed with erythrocytes to allow invasion to occur. At various time points starting at 15 min, 200 μl of culture was removed, pelleted and fixed with 2% paraformaldehyde, 3% glutaraldehyde in 0.1 M phosphate buffer (Karnovsky's fixative). After a minimum of 1 h of fixation, the cells were pelleted and suspended in 1 ml PBS to adjust the haematocrit to 0.6%. The cells were subsequently viewed using poly-L-lysine-coated 6-well Ibidi μ-slides as described above using a Nikon Ti-E inverted microscope.

## Video microscopy of invasion and amoeboid formation

Parasites were tightly synchronized using the Percoll-reinvasion-Sorbitol strategy described above and subsequently treated with ML10 to inhibit egress. Late-stage parasites were isolated on a Percoll cushion, suspended in 30 ml cRPMI containing 25 nM ML10 and incubated at 37 °C. When most parasites appeared arrested, 5 ml of the purified schizont suspension was pelleted and resuspended in warm cRPMI containing erythrocytes at a haematocrit of 3%. The parasitemia was adjusted to approximately 10% and the haematocrit was adjusted to 0.6% using warm cRPMI. 200 μl was loaded into a poly-L-lysine coated μ-Slide VI 0.4 (Ibidi) contained in a small warm bead bath. The slides were transferred to a Nikon Ti-E inverted microscope housed in a pre-warmed 37 °C chamber with an atmosphere of 5% $CO_2$. Egress, invasion of the erythrocyte by the released merozoites and the establishment of the parasites in the erythrocyte were imaged using a 100× oil immersion objective and an ORCA-Flash 4.0 CMOS camera (Hamamatsu), at a rate of 1 frame/s for 1 h. Videos were acquired and processed using Nikon NIS-Elements software.

## Immunofluorescence microscopy

Schizonts, either treated with DMSO or rapamycin as described above and subsequently purified on a Percoll cushion, were allowed to invade erythrocytes for 1–6 h and were then fixed in a solution containing 8% paraformaldehyde, 0.01% glutaraldehyde in PBS for 1 h at room temperature (Tonkin et al, 2004). The fixed parasites were permeabilized with 0.1% Triton X-100 in PBS for 10 min at room temperature, blocked with 3% BSA in PBS overnight at 4 °C and then stained with mouse anti-RESA antibodies (1:500, obtained from the Antibody Facility at The Walter and Eliza Hall Institute of

Medical Research; Reagents Table) and rabbit anti-EXP2 antibodies (1:1000, a kind gift of Paul Gilson) overnight at 4 °C. Parasites were washed three times with PBS and incubated at room temperature in the presence of fluorescently labelled secondary antibodies (1:10,000; labels used were Alexa Fluor 488 and Alexa Fluor 568 (Reagents Table)), wheat germ agglutinin (WGA)-Alexa Fluor 647 (2 μg/mL) (Reagents Table) and Hoechst 33342 (5 μg/mL) for an additional 30 min. Following three washes with PBS, 1.5 μL of parasites suspension was placed on a polyethyleneimine-coated glass slide, mixed with 1.5 μL of Vectashield anti-fade mounting medium (Reagents Table), and covered with a cover glass sealed with nail polish. Parasites were imaged on a Zeiss LSM 880 confocal microscope controlled by Zen Black version 2.3 software. The emission wavelengths used were 462 nm (Hoechst), 552 nm (Alexa 488), 633 nm (Alexa 568) and 693 nm (WGA-Alexa Fluor 647). FIJI software (Reagents Table) was used to separate and crop the images and adjust the brightness and contrast.

The proportion of RESA exported to the erythrocyte membrane was analysed as follows. Immunofluorescence microscopy images were acquired as described above and then using the WGA image, the erythrocyte surface was delineated with the FIJI threshold tool, generating a mask. This mask was then applied to the corresponding RESA image to segregate regions where RESA is exported to the erythrocyte from regions where RESA is not exported (retained in the parasite). The 'clear outside' and 'clear' functions were used to isolate these regions, respectively. The area measurement tool was then employed to quantify the proportion of exported RESA.

## Serial block-face scanning electron microscopy (SBF-SEM)

To obtain 3D models of intraerythrocytic parasites Infected erythrocytes were prepared twice for SBF-SEM. In one experiment (Experiment 1) PV6-diCre parasites were synchronized and treated with rapamycin as described above. Compound 2 was then added (1 μM final concentration) to prevent egress. When most parasites appeared arrested, the parasites were pelleted at 2000 × $g$ and suspended in warm cRPMI to remove Compound 2 and the culture was divided over four flasks. At the indicated time, a sample was removed from the cultures and immediately fixed in Karnovsky fixative and stored at 4 °C. After 40 min, Compound 2 was added to the cultures to prevent further egress and keep the parasites in a narrow window of invasion. The cultures were subsequently pelleted at 600 × $g$ and suspended in molten 1% low-melting point agarose in 0.1 M phosphate buffer, pH 7.4 and placed in a microcentrifuge tube. The samples were stored at 4 °C and prepared for microscopy as follows. Fixed agarose blocks containing cultures were washed three times for 10 min in 0.1 M phosphate buffer pH 7.4. The samples were then osmicated for 2 h in 2% aqueous osmium tetroxide. The samples were then washed three times for 10 min in distilled water and dehydrated by three 10-min washes in 30%, 50%, 70%, 90%, and 100% ethanol. The final dehydration step involved 100% dry acetone (2 times 30 min). The blocks were infiltrated overnight with 50% Durcapan resin in dry acetone before being transferred to 100% Durcapan resin for a further 12 h after which time they were cured for 24 h at 60 °C. Pieces of resin blocks containing samples were trimmed and mounted onto aluminium pins using conductive epoxy glue and silver dag, and then sputter coated with a layer (10–13 nm) of gold in an Agar Auto Sputter

Coater (Agar Scientific). Before SBF-SEM imaging, ultrathin sections microscope (JEOL) using a OneView 16-megapixel camera (Gatan-Ametek), to verify sample quality. Samples were then imaged in a Merlin VP compact high-resolution scanning electron microscope (Zeiss) equipped with a 3View 2XP stage (Gatan-Ametek) and an OnPoint backscattered electron detector (Gatan-Ametek). Image series were acquired under high vacuum, using a focal charge compensation device (Zeiss) set to 100% output. The following imaging conditions were used: (1) 1.8 kV, 20 μm aperture, 5 nm pixel size, 1–3 μs pixel time, 70 nm section thickness. SBF-SEM data were processed using the IMOD software package (Kremer et al, 1996). Briefly, image stacks were assembled, corrected (for z scaling and orientation) and aligned using eTOMO. Parasites and erythrocytes were segmented manually using 3dmod (part of the IMOD software package; Reagents Table). Movies of data and models were produced using a combination of 3dmod and Fiji (Schindelin et al, 2012). Volume, surface area and length measurements for different models were obtained in 3dmod. Statistical analysis of the data was performed using GraphPad Prism v10.

In the other experiment (Experiment 2) PV6-diCre parasites were synchronized and then treated with rapamycin approximately 30 h later. The parasites were harvested when most of them had progressed to the next cycle. The parasites were then fixed by adding 8% formaldehyde in 0.2 M Sorensen's phosphate buffer (pH 7.4) in a 1:1 ratio with the growth medium, followed by fixation in 2.5% glutaraldehyde/4% formaldehyde in 0.1 M phosphate buffer for 30 min. Next, the parasites were pelleted at $300 \times g$, embedded in 4% agarose, and cut into 1 mm$^3$ blocks on ice, and washed in PB. The parasite-agarose blocks were then prepared using a modified NCMIR protocol (Deerinck et al, 2010) by being post-fixed in 2% osmium tetroxide/1.5% potassium ferrocyanide for 1 h, incubated in 1% w/v thiocarbohydrazide for 20 min before a second staining with 2% osmium tetroxide for 30 min, then incubated overnight in 1% aqueous uranyl acetate at 4 °C. The blocks were stained with Walton's lead aspartate for 30 min at 60 °C and dehydrated through an ethanol series on ice, incubated in propylene oxide, followed by overnight incubations in a 1:1 propylene oxide/Durcupan resin mixture and twice in Durcupan resin, and embedded in Durcupan resin according to the manufacturer's instructions (TAAB Laboratories Equipment Ltd.). Images were collected using a 3View 2XP system (Gatan Inc.) mounted on a Sigma VP Scanning Electron Microscope (Carl Zeiss). Images were collected at 1.8 kV using the high-current setting, with a 20 μm aperture, at 8 Pa chamber pressure, 50 nm cutting thickness, and a pixel size of 2.5 × 2.5 nm with a 1 μs dwell time.

## Surface area analysis of erythrocytes using ImageStream

Parasites were prepared for ImageStream analysis by synchronizing parasite invasion using Percoll cushions and ML10 as described above. Parasites were washed and allowed to invade erythrocytes for 30 min and 90 min. The cells were pelleted at $2000 \times g$ and suspended in 1% glutaraldehyde in PBS at a haematocrit of 2%. Subsequently, the cells were washed twice with PBS and then suspended in PBS containing 1% Albumax at a haematocrit of 1%. Imaging flow cytometry was performed using ImageStream X Mark II (AMNIS) as described previously (Roussel et al, 2021). Briefly, infected erythrocytes were identified by labelling cells with SYBR

Green for 1 h followed by a wash with PBS, which allowed uninfected erythrocytes, erythrocytes infected with ring-stage parasites and erythrocytes infected with schizonts to be differentiated—for an example of a histogram of the SYBR Green staining and representative images of the parasites in the ring and schizont fraction, see Appendix Fig. S4). Images to determine erythrocyte dimensions and morphology were obtained using brightfield images (60X magnification) and subsequently processed using IDEAS v6.2 software (AMNIS). Focused cells and single cells were respectively selected using the features gradient RMS and aspect ratio versus area. Front views were selected and analysed by the mask "Object" and the features of circularity, perimeter and area. At least 6000 front views of focused single erythrocytes were analysed for each sample.

## Immunoblotting

Parasites were isolated on a Percoll gradient, washed several times with RPMI without Albumax and subsequently suspended in SDS-PAGE buffer at a concentration of 10$^5$ parasites/ul. Parasite lysates representing $6 \times 10^5$ parasites were separated by SDS-PAGE, transferred to nitrocellulose using a BioRad Turboblot, blocked using PBST containing 5% milk and probed with rabbit anti-PV6 (1:1000) (Hill et al, 2016) or anti-HA (3F11, 1:2000; Merck) Antibodies were visualized by incubating the blots with the appropriate HRP-linked anti-rabbit (PV6) or anti-rat (HA) secondary antibodies and developing using Clarity ECL Western blotting substrates (Bio Rad). Blots were imaged using a ChemiDoc (BioRad).

## Microsphiltration

Wild-type parasites and parasites lacking PV6 were allowed to invade erythrocytes for 4 h in cRPMI and then washed and resuspended at 1% haematocrit in PBS containing 1% Albumax (Life Technologies). The retention rate was evaluated by microsphiltration (microsphere filtration) assay in 96 wells microplates as previously described (Henry et al, 2022). The microsphiltration assay uses layers of metallic beads with different sizes to mimic the filtration of erythrocyte in the red pulp of the spleen (Deplaine et al, 2011). The retention of infected erythrocytes was analysed by flow cytometry (BD FacsCantoII, BD Biosciences) comparing parasitemia of upstream versus downstream samples stained by Hoechst DNA dye (Appendix Fig. S5). For each microplate, 95% of healthy blood donor (Etablissement Français du Sang, Lille, France) stained with carboxyfluorescein diacetate succinyl ester (CFSE) was mixed either with 5% of unstained healthy blood donor (RBC, negative control) or with 5% of 1% glutaraldehyde-fixed unstained healthy blood donor (rigid RBCs, positive control). The results from the experiment were analysed when retention rates of <10% for the negative control and >90% for the positive control were achieved.

## Perfusion of ex vivo spleen

Parasites were synchronized and treated with DMSO or rapamycin as described above. One ml pellet of the DMSO-treated culture was stained with Cell Trace Violet according to manufacturer's instructions and mixed with 1 ml of the rapamycin-treated culture that had been stained with stained with Cell Trace Far Red. The

mixture was diluted with 18 mL of healthy erythrocytes. Parasites DNA was stained with SybrGreen for the evaluation of parasitemia. The 20 ml sample was then suspended at 10% haematocrit in Krebs-albumin solution. The healthy human spleen was obtained from a patient who underwent a distal splenopancreatectomy for pancreatic disease as described previously (Buffet et al, 2006). The splenic artery was cannulated through a catheter before being connected to the perfusion platform. The spleen was co-perfused with the mixed erythrocyte population for about 60 min through the spleen at 37 °C. Samples were retrieved from a reservoir in the circuit at different time points and evaluated by flow cytometry analysis to determine the persistence of each subpopulation of infected erythrocytes. The parasitemia of each subpopulation was normalized at 100% from the start of perfusion and monitored during the time of perfusion. Owing to the scarcity of human spleens available for experimentation, this experiment was performed only once.

### Ethics

The experiments using the ex vivo human spleen are part of the "SPLEENVIVO PROTOCOL" registered by the Comité de Protection des Personnes (CPP), Île de France and approved by the IRB with the following number: 2015-02-05 MS2 DC. For the spleen used in this study, the subject signed a consent form prior to spleen retrieval and use for experiments.

## Data availability

No primary datasets have been generated and deposited.

The source data of this paper are collected in the following database record: biostudies:S-SCDT-10_1038-S44319-025-00435-3.

## Peer review information

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

## Acknowledgements

The work in this study was supported in part by a Career Development Award from the Medical Research Council to CvO (MR/R008485/1), by funding from University of Paris City and Institut National de la Santé Et de la Recherche Médicale (INSERM) to CR, PAB and PAN and by funding to MRGR, MJB and LMC from the Francis Crick Institute (https://www.crick.ac.uk/), which receives its core funding from Cancer Research UK (CC2129), the UK Medical Research Council (CC2129) and the Wellcome Trust (CC2129). The authors thank James Thomas (LSHTM) for advice on live imaging of parasite invasion, Don van Schalkwyk (LSHTM) for advice on live-dead staining of parasites, Juliana Chung (Université Paris Cité and Université des Antilles) for assistance with parasite culture and Paul Gilson (Burnet Institute) for sharing the EXP2 antibody. The authors acknowledge the facilities and the scientific and technical assistance of the LSHTM Wolfson Cell Biology Facility, with specific thanks to Liz McCarthy and Chris Chiu.

## Author contributions

**Aline Fréville**: Conceptualization; Formal analysis; Investigation; Visualization; Methodology; Writing—original draft; Writing—review and editing. **Flavia Moreira-Leite**: Methodology. **Camille Roussel**: Investigation; Visualization; Methodology. **Matthew R G Russell**: Investigation; Methodology. **Aurelie Fricot**: Investigation. **Valentine Carret**: Investigation. **Abdoulaye Sissoko**: Investigation. **Matthew J Hayes**: Investigation; Methodology; Writing—review and editing. **Aissatou Bailo Diallo**: Investigation. **Nicole Cristine Kerkhoven**: Investigation. **Margarida Ressurreição**: Investigation. **Safi Dokmak**: Resources. **Michael J Blackman**: Funding acquisition; Writing—review and editing. **Lucy M Collinson**: Supervision; Funding acquisition. **Pierre A Buffet**: Funding acquisition; Writing—review and editing. **Sue Vaughan**: Funding acquisition. **Papa Alioune Ndour**: Investigation; Visualization; Writing—review and editing. **Christiaan van Ooij**: Conceptualization; Formal analysis; Supervision; Funding acquisition; Investigation; Visualization; Methodology; Writing—original draft; Project administration; Writing—review and editing.

Source data underlying figure panels in this paper may have individual authorship assigned. Where available, figure panel/source data authorship is listed in the following database record: biostudies:S-SCDT-10_1038-S44319-025-00435-3.

## Disclosure and competing interests statement

The authors declare no competing interests.

# Expanded View Figures

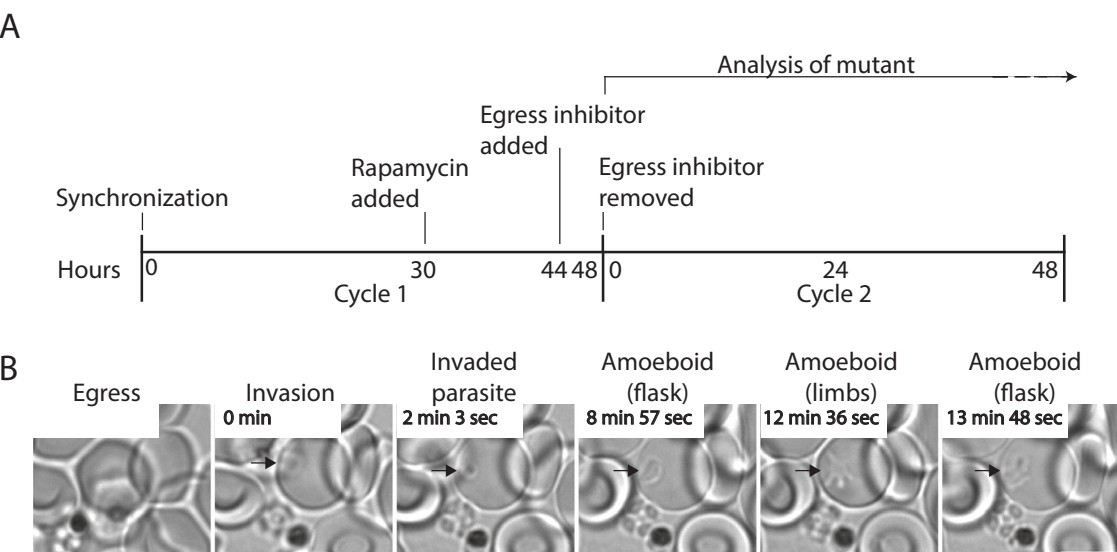

**Figure EV1. Phenotypic analyses of parasites lacking PV6.**

(**A**) Schematic representation of the protocol used to analyse the phenotype of the *P. falciparum* PV6 DiCre parasites. Synchronized parasites were treated with a minimum of 10 nM rapamycin (or an equivalent volume of DMSO) for 1 h at the trophozoite stage (~30 h post-invasion) in the first cycle. Close to the end of the first cycle, an egress inhibitor (either Compound 2 or ML10) was added to arrest the parasites at a very late schizont stage. When most of the parasites appeared arrested, the egress inhibitor was removed to initiate a round of synchronized invasion, starting cycle 2, allowing for observation of the phenotype of PV6 over time. (**B**) Live-cell imaging of amoeboid formation. The frame rate was set at one frame/second, time starting from invasion is indicated in each panel. The arrows indicate the parasite. Scale bar represents 5 μm.

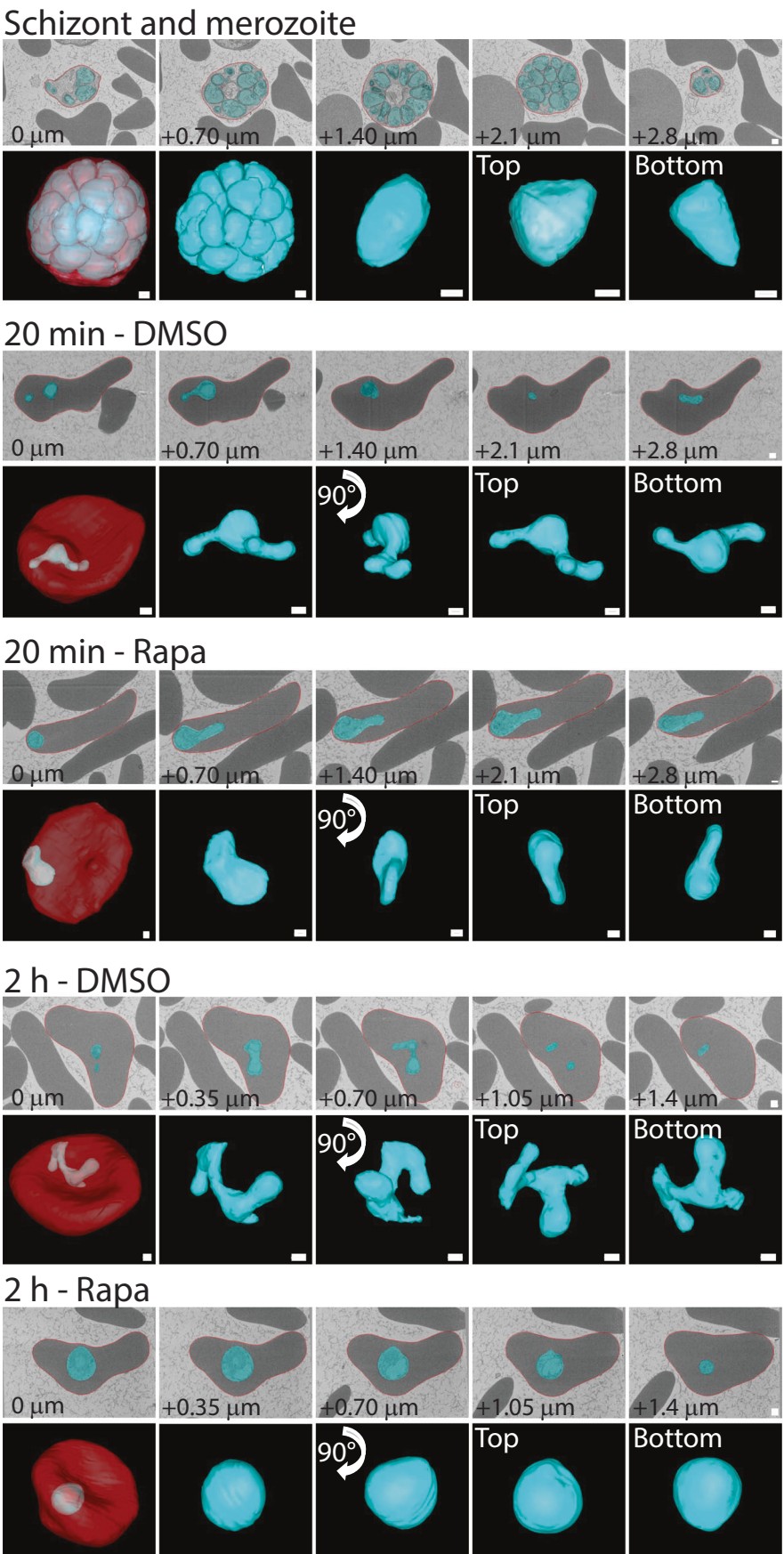

◀ **Figure EV2.  Additional three-dimensional views of the models presented in Fig. 3.**

Three-dimensional models from SBF-SEM data of the parasites presented in Fig. 3, illustrating the shape of the parasite (cyan) and its positioning within the erythrocyte (red) and additional SEM-SBF sections. The scale bars represent 500 nm.

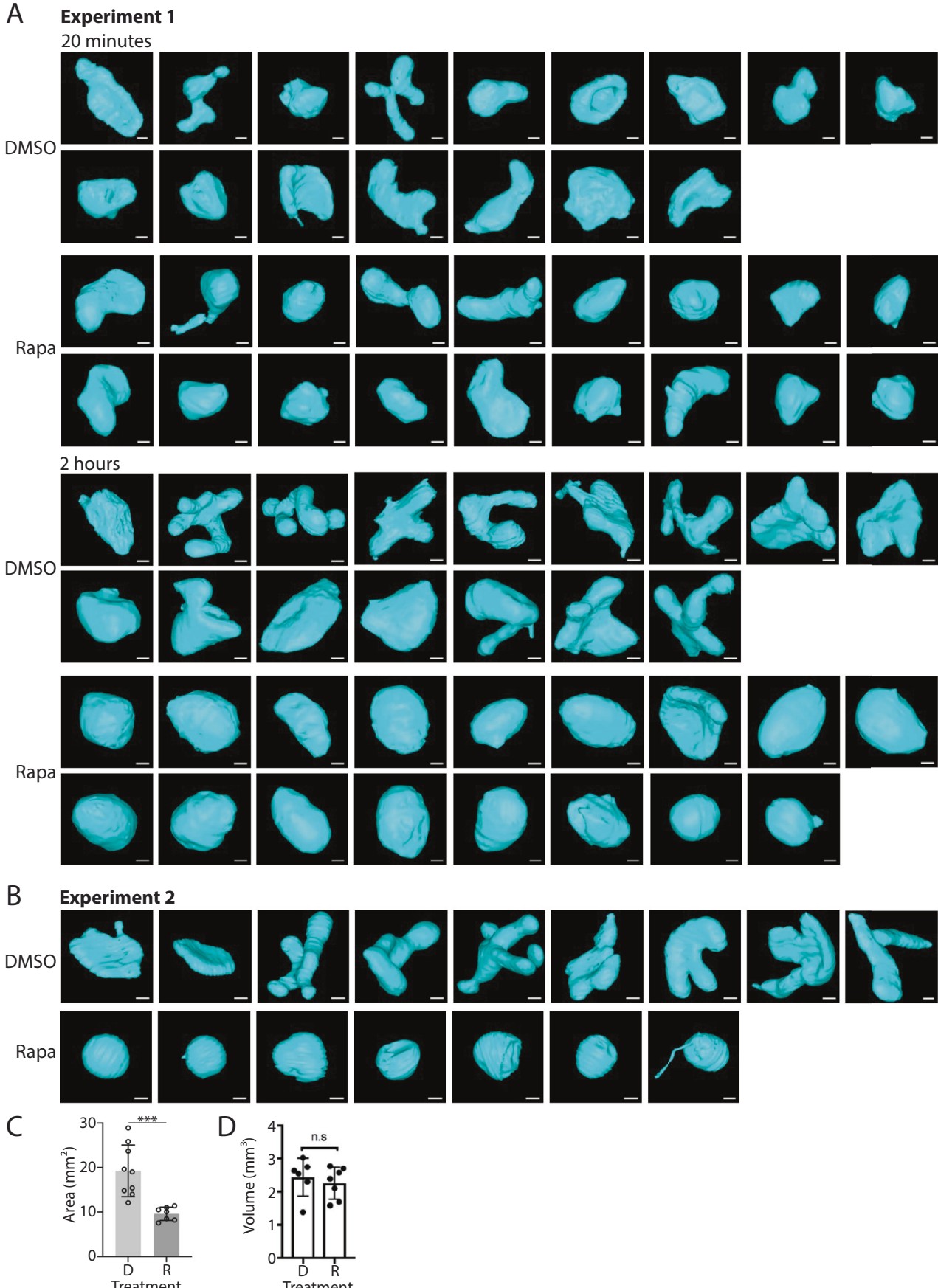

**Figure EV3.   Three-dimensional models of PfBLD529 parasites treated with DMSO or rapamycin.**

(A) These models were obtained in the same experiment as those presented in Fig. 3 and Fig. EV2. The invasion of these parasites was carefully synchronized using an egress inhibitor to allow the development of the parasites to be followed over time. Time indicated refers to the time after removal of egress inhibitor. The scale bars represent 500 nm. (B) Three-dimensional models from SBF-SEM data of DMSO-treated and rapamycin-treated parasites for which invasion was not synchronized. The parasites had been synchronized at the start of cycle 1 (Fig. EV1) and were allowed to progress to the next cycle without further synchronization. The scale bars represent 500 nm. (C) Surface area of the parasites shown in panel B; D-DMSO, R-rapamycin. Error bars $+/-$ SD. Data represent the measurement of at least 7 parasites. The Mann–Whitney U test was performed for statistical analysis (***$P < 0.001$, $P = 0.0002$). (D) Volume of the parasites shown in (B). Error bars $+/-$ SD. Data represent the measurement of at least 7 parasites. The Mann–Whitney U test was performed for statistical analysis. No significance difference was detected between the wild-type parasites and the parasites lacking PV6.

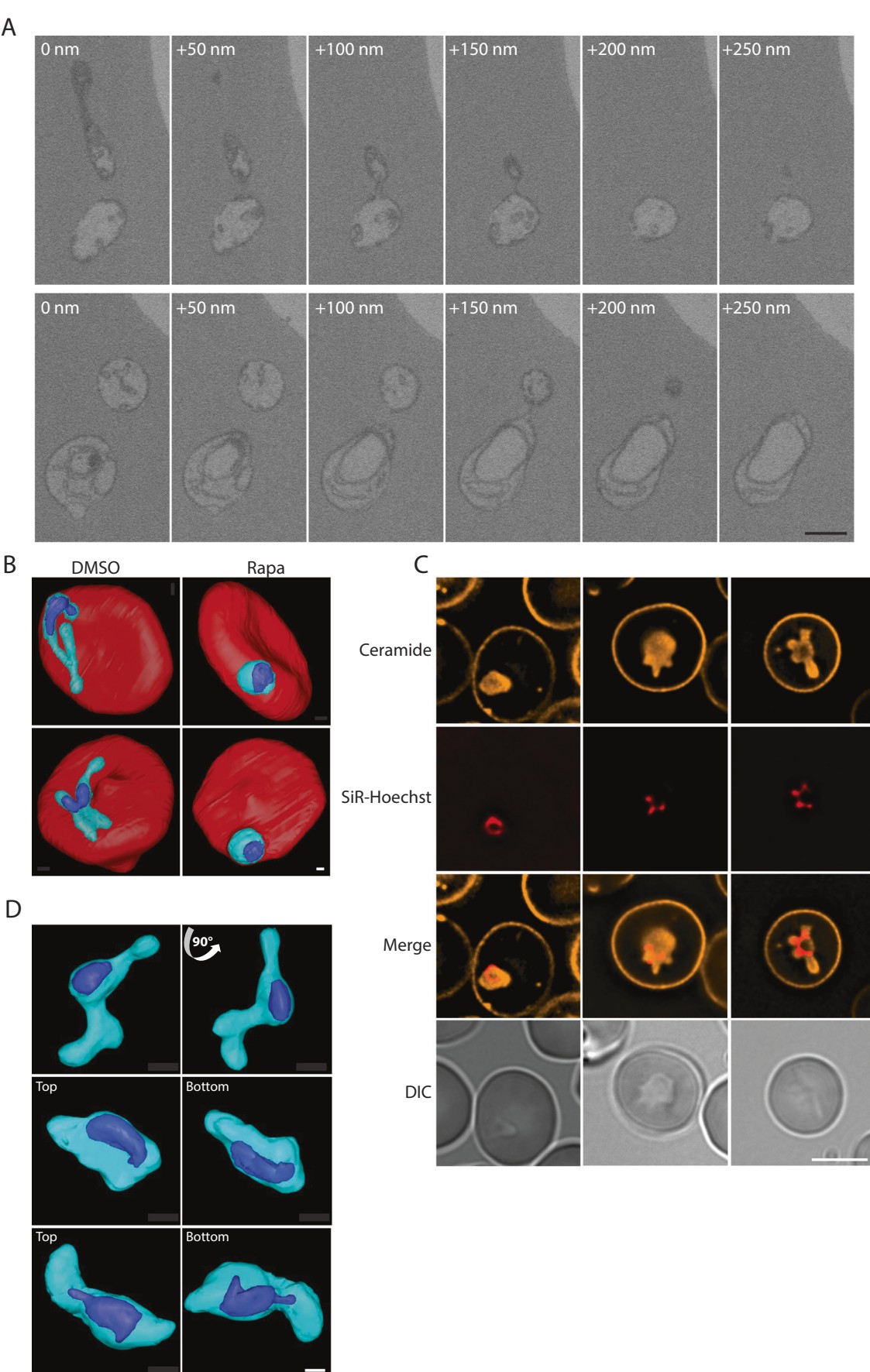

◀ **Figure EV4. Nuclear morphology in wild-type amoeboid-shaped parasites and parasites lacking PV6.**

(A) Consecutive SBF-SEM sections, showing the thin connections between limbs of two different parasites. Numbers in the upper left-hand corner of the panels indicate distance between the sections and the scale bar represents 500 nm. (B) Three-dimensional models from SBF-SEM data of erythrocytes infected with DMSO-treated and rapamycin-treated parasites 2 h after removal of egress inhibitor. Erythrocytes (red), parasite (cyan) and the nucleus (blue) are highlighted. The scale bar represents 500 nm. (C) Live-cell fluorescence imaging of infected erythrocytes labelled with C5-Bodipy-ceramide (orange) and SiR-Hoechst (red) 2 h after removal of egress inhibitor. Note that in the merged imaged, the parasite and the SiR-Hoechst do not overlap perfectly owing to the movement of the parasite during the acquisition of the images. The scale bar represents 5 μm. (D) Three-dimensional model showing the nucleus (blue) in wild-type parasites (magenta) 20 min after removal of egress inhibitor. Scale bar represents 500 nm.

A

DMSO · Rapa

B

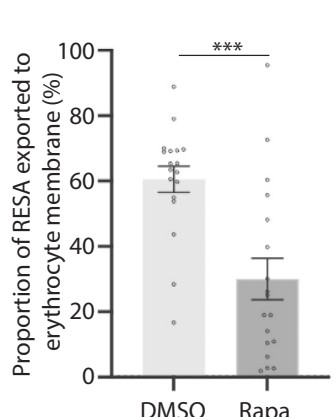

**Figure EV5. Additional SBF-SEM images of wild-type parasites and parasites lacking PV6.**

(A) Individual SBF-SEM sections of different erythrocytes infected with wild-type PfBLD529 parasites (left; DMSO) and PfBLD529 parasites lacking PV6 (right; Rapa). In each case, the left-hand panel shows a close view of the parasite and the right-hand panel shows the entire infected erythrocyte. Note the accumulation of membranous whorls next to the parasites lacking PV6. The close views of the parasites in the top row are also shown in Fig. 5. Scale bars represent 500 nm. (B) Analysis of export of RESA in erythrocytes infected with DMSO-treated and rapamycin-treated parasites. Samples prepared for IFA were imaged as described in the Methods section. The amount of RESA exported to the erythrocyte was measured for 18 parasites obtained from two independent experiments. Error bars $+/-$ SD. The Mann–Whitney U test was performed for statistical analysis (***$P < 0.001$ ($P = 0.0005$)).

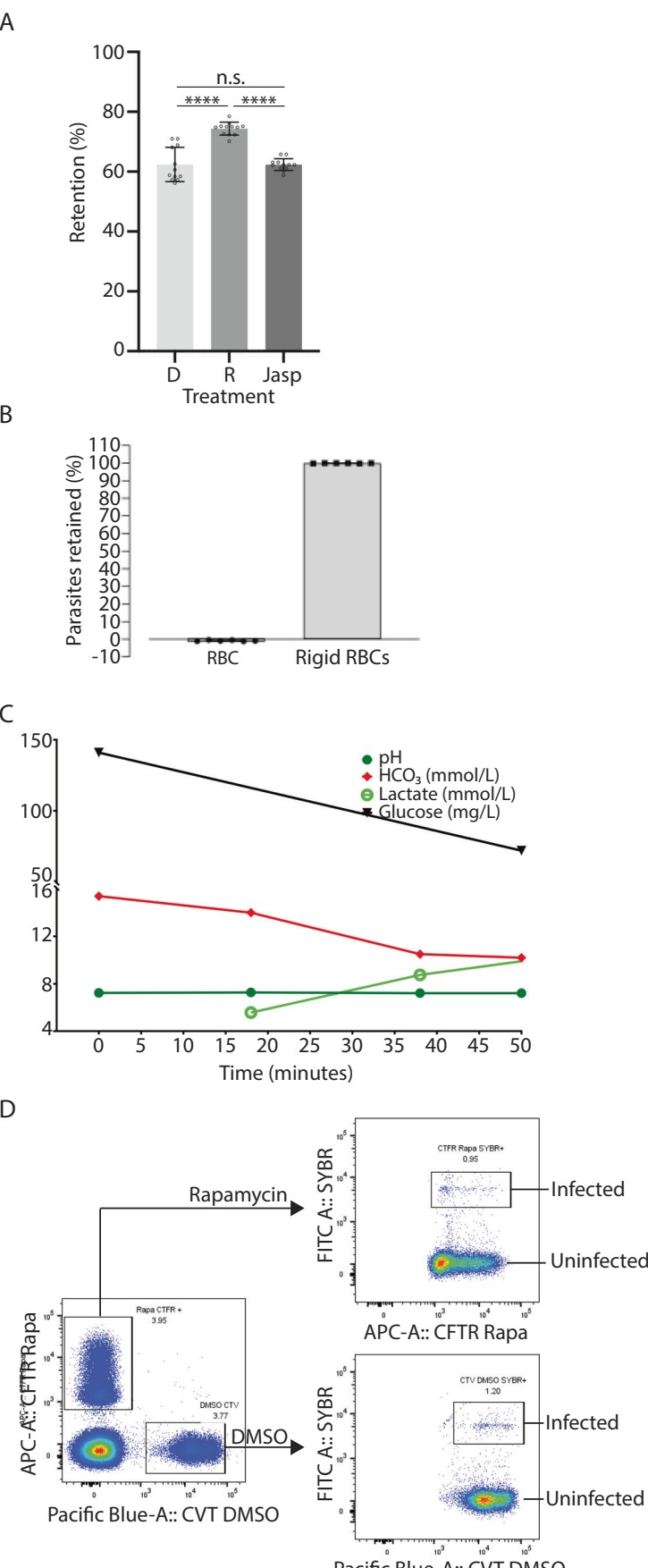

◀ **Figure EV6. Analysis of samples used for microsphiltration and ex vivo spleen perfusion.**

(A) Retention of PV6-diCre parasites cultures treated with DMSO, rapamycin or jasplakinolide. Data represent 2 biological replicates with a minimum of 6 technical replicates. Error bars: $+/-$ SD. The Mann–Whitney U test was performed for statistical analysis: ns, not significant; ****$P < 0.0001$. (B) Retention of untreated erythrocytes (RBCs) and fixed erythrocytes (rigid RBCs) on the columns used for the microsphiltation analysis of erythrocyte infected with wild-type parasites or parasites lacking PV6. Data represents 1 biological replicate with 6 technical replicates. (C) Physiological state of ex vivo spleen during the course of the perfusion with infected erythrocytes. The pH and levels of bicarbonate, lactate and glucose were measured periodically to ensure the proper functioning and viability of the spleen. (D) Gating strategy used for cytometric analysis of blood passaged through ex vivo spleen.

