## [Peer Review File · EMBO Reports]

Malaria parasites undergo a rapid and extensive metamorphosis after invasion of the host erythrocyte

Aline Fréville, Flavia Moreira-Leite, Camille Roussel, Matthew Russell, Aurelie Fricot, Valentine Carret, Abdoulaye Sissoko, Matthew Hayes, Aissatou Bailo DIALLO, Nicole Kerkhoven, Margarida Ressurreição, Safi Dokmak, Michael Blackman, Lucy Collinson, Pierre Buffet, Sue Vaughan, Papa Alioune Ndour, and Christiaan van Ooij

Corresponding author(s): Christiaan van Ooij (c.van.ooij@keele.ac.uk)

Review Timeline:

Submission Date:	20th Sep 24
Editorial Decision:	11th Oct 24
Revision Received:	13th Jan 25
Editorial Decision:	5th Feb 25
Revision Received:	21st Feb 25
Accepted:	25th Feb 25

Editor: Achim Breiling

Transaction Report:

Dear Dr. Van Ooij,

Thank you for the submission of your manuscript to EMBO reports. I have now received the reports from the three referees that were asked to evaluate your study, which can be found at the end of this email.

As you will see, the referees find the study very interesting. Nevertheless, they have several comments, concerns, and suggestions, indicating that a major revision of the manuscript is necessary to allow publication of the study in EMBO reports. As the reports are below, and all the concerns need to be addressed, I will not detail them further here.

Given the constructive referee comments, I would like to invite you to revise your manuscript with the understanding that the concerns of the referees must be addressed in the revised manuscript and in a detailed point-by-point response. Acceptance of your manuscript will depend on a positive outcome of a second round of review. It is EMBO reports policy to allow a single round of revision only and acceptance of the manuscript will therefore depend on the completeness of your responses included in the next, final version of the manuscript.

- 1) a .docx formatted version of the final manuscript text (including legends for main figures, EV figures and tables), but without the figures included. Figure legends should be compiled at the end of the manuscript text.
- 2) individual production quality figure files as .eps, .tif, .jpg (one file per figure), of main figures and EV figures. Please upload these as separate, individual files upon re-submission.

- 4) a complete author checklist, which you can download from our author guidelines (<https://www.embopress.org/page/journal/14693178/authorguide>). Please insert page numbers in the checklist to indicate where the requested information can be found in the manuscript. The completed author checklist will also be part of the RPF.

- 5) that primary datasets produced in this study (e.g. RNA-seq, ChIP-seq, structural and array data) are deposited in an

appropriate public database. If no primary datasets have been deposited, please also state this in a dedicated section (e.g. 'No primary datasets have been generated and deposited'), see below.

The accession numbers and database should be listed in a formal "Data Availability" section that follows the model below. This is now mandatory (like the COI statement). Please note that the Data Availability Section is restricted to new primary data that are part of this study. This section is mandatory. As indicated above, if no primary datasets have been deposited, please state this in this section

Data availability

8) Regarding data quantification and statistics, please make sure that the number "n" for how many independent experiments were performed, their nature (biological versus technical replicates), the bars and error bars (e.g. SEM, SD) and the test used to calculate p-values is indicated in the respective figure legends (also for EV and Appendix figures). Please also check that all the p-values are explained in the legend, and that these fit to those shown in the figure. Please provide statistical testing where applicable. Please avoid the phrase 'independent experiment', but clearly state if these were biological or technical replicates. Please also indicate (e.g. with n.s.) if testing was performed, but the differences are not significant. In case n=2, please show the data as separate datapoints without error bars and statistics. See also: <http://www.embopress.org/page/journal/14693178/authorguide#statisticalanalysis>

9) Please add scale bars of similar style and thickness to microscopic images, using clearly visible black or white bars (depending on the background). Please place these in the lower right corner of the images themselves. Please do not write on or near the bars in the image but define the size in the respective figure legend.

10) Please also note our reference format:

12) We now use CRedit to specify the contributions of each author in the journal submission system. CRedit replaces the author contribution section. Please use the free text box to provide more detailed descriptions and do NOT provide your final manuscript text file with an author contributions section. See also our guide to authors: <https://www.embopress.org/page/journal/14693178/authorguide#authorshipguidelines>

13) All Materials and Methods need to be described in the main text using our 'Structured Methods' format, which is required for

all research articles. According to this format, the Methods section should include a Reagents and Tools Table (listing key reagents, experimental models, software, and relevant equipment and including their sources and relevant identifiers), uploaded as separate file, followed by a Methods section in which we encourage the authors to describe their methods using a step-by-step protocol format with bullet points, to facilitate the adoption of the methodologies across labs. More information on how to adhere to this format as well as downloadable templates (.doc) for the Reagents and Tools Table can be found in our author guidelines (section 'Structured Methods'):

14) Please add up to 5 keywords to the manuscript and order the manuscript sections like this, using these names: Title page - Abstract - Keywords - Introduction - Results - Discussion - Methods - Data availability section - Acknowledgements (including funding information) - Disclosure and Competing Interests Statement - References - Figure legends - Expanded View Figure legends

15) Please provide all methods information in the main manuscript text file. We do not allow supplementary methods.

16) Please name and upload the movie files using the name 'Movie EVx' and update the callouts accordingly. Please provide the legends/descriptions for the movies as text file that is ZIPed together with the respective movie file and uploaded. Then please remove the legends for the movies from the main article file.

17) Please make sure that all the funding information is also entered into the online submission system and that it is complete and similar to the one in the acknowledgement section of the manuscript text file.

I look forward to seeing a revised form of your manuscript when it is ready.

Yours sincerely,

Referee #1:

Upon RBC infection, *Plasmodium falciparum* parasites convert from an invasive merozoite to a vacuole-dwelling intracellular ameboid form in the early ring stage. In this study, Freville et al dissect this morphological transition using a conditional mutant of PV6, a START-domain protein thought to mediate lipid transfer to the PVM. Parasites lacking PV6 do not form ameboids and also fail to increase PV/parasite area during the first 30 minutes of infection, confirming recent observations with PV6 inhibitors. A high point of the paper are high resolution parasite volume and surface area measurements from SBF-SEM showing the transition from merozoite to ring stage occurs in two steps with only the second step requiring PV6. Finally, the authors show that the decrease in RBC surface area upon infection (resulting from PVM formation from the host plasma membrane) is more substantial with the PV6 knockout. Additionally, RBCs infected with the PV6 mutant experience greater retention on a microfiltration column and increased filtration when perfused through an ex vivo human spleen.

Overall this is an interesting study that reveals new steps in the transition to intraerythrocytic development and suggests proper transition to the early ring stage is important to avoid splenic clearance. However, I have some concerns regarding some of the conclusions outlined below.

Major Comments:

-PV lumen membrane whorls are seen in the of the PV6 mutant. How frequent was this observed? Does this indicate the volume

of the PV lumen increased in the mutant? Can the SBF-SEM data be segmented to measure the volume of the PV or to measure the parasite surface area (parasite plasma membrane) separately from the PVM to gain more insight about PV morphology and parasite size in the mutant?

-The increased volume of the Δ PV6 parasites observed by SBF-SEM is surprising given the decreased area observed by light microscopy in fixed and live parasites (Fig 2B,D) but is not discussed. Is the volume of the Δ PV6 parasites also higher than the control in the second SBF-SEM experiment shown in Supp Fig 4B-D (as observed for the first experiment at 2 hr time point in Fig 3D)? I'm confused how this can occur if PVM/PPM expansion is not occurring in the absence of PV6 - some additional commentary is needed.

-Fig 5C: the authors should provide quantification and a statistical test to support the claim that RESA is not exported in the PV6 mutant.

-Does increased retention on the microfiltration column result from the lack of amoeboid morphology (due to increase parasite flexibility) or the decreased RBC surface area? Assuming the RBC area is not affected by jasplakinolide treatment, this could be tested by measuring retention of jasplakinolide treated parasites on the column since this prevents amoeboid morphology but not growth.

-The authors propose the model that the PVM is formed from the host plasma membrane and then grown by addition of lipids from discharged rhoptry material or the parasite through PV6 activity. If so, it seems that PV6 would be important beyond the ring stage as well when the PVM expands substantially. However, PV6 inhibitors do not seem to impact development after the early ring stage (Dans et al, PMID 38890312 - perhaps the authors can provide additional insight with their mutant if the timing of rapamycin treatment is altered to deplete PV6 in later stages). If PV6 is not important beyond the ring stage, this would indicate that either PV6 uniquely supports very early PVM growth and a different mechanism mediates PVM growth in later stages, or that PV6 is not actually involved in PVM expansion. This should be discussed.

-The reduced surface area of RBCs infected with the PV6 mutant is surprising as the PVM is not connected with the RBC membrane and other factors would be needed outside the PV to transfer lipids to the host membrane. This is mostly contributed by the shoulder in 6A in the PV6 mutant in the range of 40-60 μm^2 . These cells are surprisingly small - does the imaging flow cytometry data provide any insight about the nature of these iRBCs? The authors should also provide the surface area differences for DMSO uRBCs vs DMSO iRBCs in 6B to show the magnitude of the effect in wild type iRBCs (decrease in iRBC surface area relative to uRBC as indicated in the text on pg 11, line 28) for comparison with the uRBC DMSO/RAP and iRBC DMSO/RAP. The methods indicate samples were taken at 30 and 90 min after removal of ML10. Which time point is shown in the figure or are the data pooled? Were there any differences between the time points? The statistical details in the legend description (pg 26, line 11) should be moved from the 6A description to the next section (6B) in the legend.

Minor Comments:

-Fig 2D-E: the grey circles are difficult to see against the darker grey bars in D Jasp and E Rapa and Jasp. Please adjust to increase the contrast.

-pg 9, lines 20-21: This sentence implies that the first expansion occurs by the time invasion is completed but the data do not resolve between the end of invasion and the first ~5 minutes of intracellular development post-invasion. Why is the first expansion assumed to occur before invasion is complete rather than after (which seems more likely to me)?

Grammar:

-pg 12, line 25: "possibly" should be "possibly"

Referee #2:

In this manuscript Fréville et al present data on the morphological transformations of *P. falciparum* parasites immediately after they invade a new red blood cell (RBC). Very little research has so far focused on this life cycle phase and this manuscript is therefore of high interest and novelty. One part of this study describes the morphological alterations in WT parasites. The other part of the manuscript uses a conditional mutant of the gene encoding PV6, a lipid transfer protein. The mutant parasites invade RBCs, but do not make amoeboid cells and have a smaller surface area than normal rings. The authors use these cells to test what happens if the transformation to the amoeboid parasite does not occur. Interestingly they find evidence that the PV6 mutant is more prone to splenic clearance. A further interesting finding is that the PV6 mutant does not seem to export proteins.

This is a very interesting paper on a so far poorly studied part of the life cycle of the parasite. The data is clearly presented and I have only non-critical comments:

- The introduction states that PV6 mediates changes to the RBC that are needed for survival. Given the lack of export of RESA and possibly other exported proteins it is likely that this is due to a lack of protein export. Or is the idea that PV6 directly changes

the RBC?

- Fig. 2C - E. Did the PV6 mutant parasites move around in the RBC like is known from WT rings? The EMs in the manuscript indicate the mutant cells may be stuck in the periphery of the RBC. Could this contribute to the failure of proper development into a spherical ring? Determining whether the mutant is arrested in position or not would be easy (or probably was already done along the experiments shown in Fig. 2C-E) and might give further indications what is wrong with the PV6 mutant.
- L1-6/P8: the argument makes sense that PV6 is needed to increase surface area but not necessarily to develop an amöboid shape. However, the last sentence of this section is then somewhat confusing as it states PV6 is needed to progress into the amöboid form. Maybe it might be more accurate to turn this sentence around so that it states that PV6 is needed for the parasite to increase in surface area and that this in turn is a prerequisite to adopt an amöboid shape.
- same section and also other parts in the paper: 'PV6 is required to progress to the amöboid form'. To me it sometimes seems to imply that the amöboid form is equivalent to a life cycle stage the parasite has to reach. However, amöboidity is not a stage the parasite has to progress to but a state a typical ring can (maybe facultatively) adopt. This is not to say an amöboid shape is not a good indicator that a cell has reached the typical ring stage (i.e. completed the metamorphosis), but it might be good to use wording that makes clear it is a state, rather than a stage. It is even possible some rings never show an amöboid shape and remain in the extended spherical shape.
- Given their tiny appearance in Giemsa smears I have to admit I find it surprising that the volume of the mutant is larger than that of the WT (Fig. 3D). The surface measurement was reproduced in Fig. 4B and C. It would be nice to have the volume data for these experiments as well.
- Some support information for Fig. 6 would be nice. For instance, it was before mentioned (L31/P8) that the parasite preparations had some schizonts in them. Was the rate of schizonts in the samples that went into these experiments comparable between WT and PV6 mutant? For the experiment shown in Fig. 6A it might have been possible to distinguish stage of the cells, were for instance schizont-infected RBCs excluded? Was the input and output of the microfiltration smeared? There is a FACS plot for the spleen experiment which is a good way to determine parasitemia, but a smear gives more accurate information what exactly went in and came out.
- I am aware that there might be a confounding effect on the RBC, but maybe this is still worth testing: did the authors run the microfiltration experiment (Fig. 6C) also with jasplakinolide treated WT comparing to control and the mutants (plus/minus) jasplakinolide? This might show whether amöboidity is a contributor of the effect.
- The authors write that PV6 promotes the passage through the spleen. This sounds very active, maybe safer to say that if it is not there, spleen passage is reduced.
- The use of the semicolon in the paper was slightly confusing to me. I am not a native speaker, but maybe this could be revisited, for instance by replacing with a colon in some places.
- L12/P5 remove the bracket with ref in it.
- In the discussion it is stated that the PV6 parasites are smaller. But if I correctly understand they are smaller in surface area but larger in size (Fig. 3D)? It is important to make this distinction.
- L28/P11 insert space before This

Referee #3:

Freville et al report on the development of the human malaria parasite *Plasmodium falciparum* immediately after invasion of the host erythrocyte and the role of the phospholipid transfer protein PV6 in this process. The authors demonstrate that *Plasmodium falciparum* malaria parasites undergo multiple morphological changes soon after erythrocyte invasion and implicate this in the parasites ability to avoid splenic clearance in the host. They further demonstrate that the protein PV6 plays a role in at least one part of this process, the formation of amoeboid like rings. This manuscript is well written with only minor spelling/grammatical errors as detailed below. Previously published literature on this topic has been referenced appropriately and in context with the results presented here. Some of the interpretations, or their explanations, are a bit challenging to support and the manuscript would benefit from some more thought into how this could best be done.

Major Comments:

Page 8, lines 1-6 and Figure 2 - The interpretation around shape could be clearer here. Text states that WT parasites with jasplakinolide had shape affected but not area. Based on 2C jasplakinolide appears to differ from DMSO-treated control which is amoeboid and seems to be more spherical than Rapalog treated samples but based on 2E, it doesn't seem that shape was greatly affected compared to DMSO control.

For 2D, there is a NS difference between DMSO and Jasplakinolide, but a Sig difference between DMSO/Jas and Rapamycin treatment. This is interpreted as Jasp not affecting area.

For 2E, there is a NS difference between DMSO and Jasplakinolide, but a Sig difference between DMSO/Jas and Rapamycin treatment. This is interpreted as Jasp affecting the shape??? Should not this comparison, as for area, be a comparison with DMSO, where there is NS difference for Jasp?

Based on this data, I would interpret that Jasp treatment showed no significance difference in Area or Shape when compared to the DMSO control.

Page 12, line 1: Based on the lower erythrocyte volume for rapamycin treated PV6 loss of function mutants it is hypothesised that this protein is involved in mitigating erythrocyte volume loss. Is there any evidence that loss of PV6 leads to an increase in parasitophorous vacuole membrane volume, since this is where some erythrocyte membrane is lost. Have these measurements been done? If not, where could the membrane have gone? In the discussion, the possibility that PV6 somehow sends erythrocyte membrane back to the cell surface is put forward, though the authors acknowledge that this seems like a difficult thing for PV6 to achieve when it is located in the PV. How might this work then. Could it be at the point where the tight junction closes off so that less erythrocyte membrane is taken in during PVM formation? Is it possible that loss of PV6 selects for invasion into small erythrocytes somehow?

There seems to be two hypothesis put forward into why splenocyte models preferentially removed PV6 loss of function infected erythrocytes over those that had PV6. The first is the smaller size of the erythrocyte with loss of PV6, but it seems counter intuitive that a smaller cell in a bead based model be preferentially removed. Second is the PV6 expressing parasites can form amoeboid shapes that make them, and their erythrocytes, more flexible. Whereas loss of PV6 maintains the round merozoite state. Merozoites are much smaller than the erythrocytes they infect. Would they be big enough to impact on progression through the spleen? These models are put forward largely separately and not really discussed together. This makes it unclear which, or whether both, models are likely contributing the most. Linking these models in the discussion and discussing whether either would impact the likelihood of the other would be of benefit.

Minor Comments:

There a number of small text errors, some of which are highlighted below.

Page 5, line 12: remove '(ref)'.

Page 6, line 4: Egress an invasion are said to be precisely timed, but then this is said to happen 'approximately' 15 minutes after removal of Compound2/ML10. Suggest that the word precisely is removed as there is a probably range of several minutes with this method.

Page 6, line 12: The difference in the rate of conversion to the amoeboid form between two experiments is posited to be from observing the cells. Is it also possible that differences in the starting late stage parasite population and relative period on Compound2/ML10 during development could also impact downstream events?

Page 6, lines 21 - 32: Less amoeboid forms in Pk parasites. This is an interesting comparison, but there's no statement of the significance or further discussion of this so from that perspective, this paragraph lacks some relevance.

Page 11, line 28: gap between sentences.

Page 10, line 17: change 'can occurs' to either 'can occur' or 'occur'.

Page 13, line 5: 'further likely', perhaps change to either just 'further' or just 'likely'.

Page 16, line 5: The *P. falciparum* culture method mentions selecting for rings two different ways (precoll, sorbitol). Can this be reworded.

Page 23, lines 27 and 31: 'Error bars' instead of 'Errors bars'.

Page 24, line 5: change 'fornation'.

Page 24, lines 27 and 29: 'Error bars' instead of 'Errors bars'.

Page 25, line 19: Change 'parasites' to 'parasite', assuming singular.

Page 25, line 20 - 21: 'Live cell imaging of an infected erythrocyte stained with ceramide, the DNA dye SiR-Hoechst, the merge of the ceramide and SiR-Hoechst images and imaged using DIC'. Confusing sentence.

Page 26, lines 11 and 17: 'Errors bars' again.

Figure 1B: panels on the right depicting amoeboid forms. The 25-minute image could show a clearer amoeboid form if one is available.

The term 'wildtype' parasites is used throughout for a gene-edited line, including for DMSO treated cultures. Even though the gene has not been excised, they have still been altered so would it be more accurate to refer to these as the DMSO control? Or as the gene-name with the gene-edited parts.

We thank the referees for their time and their constructive comments. We have addressed these comments to the best of our ability and we believe that as a result of the referees' feedback, the manuscript has improved substantially. Below we address the comments point-by-point.

Referee #1:

Upon RBC infection, *Plasmodium falciparum* parasites convert from an invasive merozoite to a vacuole-dwelling intracellular ameboid form in the early ring stage. In this study, Freville et al dissect this morphological transition using a conditional mutant of PV6, a START-domain protein thought to mediate lipid transfer to the PVM. Parasites lacking PV6 do not form ameboids and also fail to increase PV/parasite area during the first 30 minutes of infection, confirming recent observations with PV6 inhibitors. A high point of the paper are high resolution parasite volume and surface area measurements from SBF-SEM showing the transition from merozoite to ring stage occurs in two steps with only the second step requiring PV6. Finally, the authors show that the decrease in RBC surface area upon infection (resulting from PVM formation from the host plasma membrane) is more substantial with the PV6 knockout. Additionally, RBCs infected with the PV6 mutant experience greater retention on a microfiltration column and increased filtration when perfused through an ex vivo human spleen.

Overall this is an interesting study that reveals new steps in the transition to intraerythrocytic development and suggests proper transition to the early ring stage is important to avoid splenic clearance. However, I have some concerns regarding some of the conclusions outlined below.

Major Comments:

-PV lumen membrane whorls are seen in the of the PV6 mutant. How frequent was this observed?

We have counted the number of parasites that display the whorls in one dataset. The results, shown in Table 1, indicate that whorls are seen in only 18% of the wildtype parasites and 60% of the mutant parasites. Furthermore, the whorls in wildtype parasites were smaller than those seen in wildtype parasites. The results are described on p11, lines 14-17 and 22-24.

Does this indicate the volume of the PV lumen increased in the mutant? Can the SBF-SEM data be segmented to measure the volume of the PV or to measure the parasite surface area (parasite plasma membrane) separately from the PVM to gain more insight about PV morphology and parasite size in the mutant?

This is a very good point. We have tried to do this, but the resolution of the images was not sufficient to do this to a level that gives reliable data.

-The increased volume of the Δ PV6 parasites observed by SBF-SEM is surprising given the decreased area observed by light microscopy in fixed and live parasites (Fig 2B,D) but is not discussed. Is the volume of the Δ PV6 parasites also higher than the control in the second SBF-SEM experiment shown in Supp Fig 4B-D (as observed for the first experiment at 2 hr time point in Fig 3D)? I'm confused how this can occur if PVM/PPM expansion is not occurring in the absence of PV6 - some additional commentary is needed.

We agree that this seems counterintuitive. Possibly the lack of export indicates that EXP2 is not functional and besides not transporting proteins also cannot transport water, trapping any contents secreted from the apical organelles in the PVM. We have added the volume measurements for the second experiment (EV3D), in which we did not detect a difference in volume, although the lower level of synchronization in that experiment was less, which may have affected the results. We have added a brief section in the discussion to address this (p15, lines 5-9)

-Fig 5C: the authors should provide quantification and a statistical test to support the claim that RESA is not exported in the PV6 mutant.

We have quantitated the amount of RESA exported in two separate experiments, and a total of 18 rings per condition, which back up our statement that a smaller proportion of RESA is exported from parasites lacking PV6 than from wild type parasites. The data are presented in Expanded View Fig 5B.

-Does increased retention on the microfiltration column result from the lack of amoeboid morphology (due to increase parasite flexibility) or the decreased RBC surface area? Assuming the RBC area is not affected by jasplakinolide treatment, this could be tested by measuring retention of jasplakinolide treated parasites on the column since this prevents amoeboid morphology but not growth.

This is a very good comment. We have added a figure (Expanded view 7A) that shows the results of a microfiltration experiments using two different strains of PV6-diCre parasites treated with jasplakinolide (see also p13, lines 9-14). The results reveal that the circular jasplakinolide-treated parasites pass through the column with the same efficiency as the untreated parasites. Although it remains unknown how flexible and malleable jasplakinolide-treated cells are, this result appears to indicate that the spherical jasplakinolide-treated parasites can pass through the column (and hence the spleen). In the text (p15, lines 18-24), we have emphasized that the decreased surface area of the erythrocyte infected with the mutant is the likely reason that it is more prone to removal on a microfiltration column and in a spleen.

-The authors propose the model that the PVM is formed from the host plasma membrane and then grown by addition of lipids from discharged rhoptry material or the parasite through PV6 activity. If so, it seems that PV6 would be important beyond the ring stage as well when the PVM expands substantially. However, PV6 inhibitors do not seem to impact development after the early ring stage (Dans et al, PMID 38890312 - perhaps the authors can provide additional insight with their mutant if the timing of rapamycin treatment is altered to deplete PV6 in later stages). If PV6 is not important beyond the ring stage, this would indicate that either PV6 uniquely

supports very early PVM growth and a different mechanism mediates PVM growth in later stages, or that PV6 is not actually involved in PVM expansion. This should be discussed.

We have added a brief mention of this in the discussion – see p14, lines 30-32, p15, lines 1-5. In our opinion, the results presented by Dans *et al* addressing the role of PV6 in the trophozoite stage are inconclusive. The late-stage parasites treated with PV6 inhibitor shown in that study appear to be smaller and more condensed than the control parasites. Preliminary data from our lab has revealed that when the diCre-PV6 parasites are treated with rapamycin during the early ring stage, producing schizonts lacking PV6, these parasites are smaller than parasites that produce PV6. We have not included these data in this manuscript as we want to maintain the focus on the merozoite-amoeboid transition and the role of PV6 in this process.

-The reduced surface area of RBCs infected with the PV6 mutant is surprising as the PVM is not connected with the RBC membrane and other factors would be needed outside the PV to transfer lipids to the host membrane. This is mostly contributed by the shoulder in 6A in the PV6 mutant in the range of 40-60 μm^2 . These cells are surprisingly small - does the imaging flow cytometry data provide any insight about the nature of these iRBCs?

In Appendix Fig S3, we provide galleries of unselected images corresponding to the infected red blood cells found in the shoulder and in the main population. The shoulder corresponds to small spherocytic and echinocytic (late stages echinocytes i.e echinocytes III and spheroechinocytes) iRBCs while the main population correspond mostly to discocytes and early stages echinocytes (Echinocytes I and II).

The extent of surface area loss of these morphologically altered RBCs matches what we previously observed in the context of RBC storage lesions (Roussel et al, Blood 2021, please see below).

B

Of note, this extent of surface area loss was shown to induce complete retention by microsphiltration and in ex vivo perfusion of human spleen (Safeukui et al, 2012; Safeukui et al, 2013; Roussel et al 2021) through loss in surface-to-volume ratio

The authors should also provide the surface area differences for DMSO uRBCs vs DMSO iRBCs in 6B to show the magnitude of the effect in wild type iRBCs (decrease in iRBC surface area relative to uRBC as indicated in the text on pg 11, line 28) for comparison with the uRBC DMSO/RAP and iRBC DMSO/RAP.

We have mentioned the results of this comparison (p16, lines 3-6) and have now added the data as Expanded View Figure 6.

The methods indicate samples were taken at 30 and 90 min after removal of ML10. Which time point is shown in the figure or are the data pooled?

These samples were taken 30 minutes after removal of ML10. This is now indicated in the legend for the figure. The data from the two experiments performed with parasites 90 minutes after removal of ML10 are shown in Table 3.

Were there any differences between the time points?

The surface area of erythrocytes infected with parasites lacking PV6 was decreased similarly compared to erythrocytes infected with parasites at 30 minutes and 90 minutes after removal of ML10. These results are shown in Table 3. We have also mentioned this in the text p12, lines 24-25.

The statistical details in the legend description (pg 26, line 11) should be moved from the 6A description to the next section (6B) in the legend.

We thank the reviewer for pointing this out and have moved the statistical detail as suggested.

Minor Comments:

-Fig 2D-E: the grey circles are difficult to see against the darker grey bars in D Jasp and E Rapa and Jasp. Please adjust to increase the contrast.

We have changed the grey circles to black throughout.

-pg 9, lines 20-21: This sentence implies that the first expansion occurs by the time invasion is completed but the data do not resolve between the end of invasion and the first ~5 minutes of intracellular development post-invasion. Why is the first expansion assumed to occur before invasion is complete rather than after (which seems more likely to me)?

The reviewer is correct that the expansion may well occur within a few minutes after invasion. This is in fact what we also think happens – the sentence was poorly worded. We have changed the sentence to reflect that the parasites could expand after invasion has been completed (p 10, line 4).

Grammar:

-pg 12, line 25: "possibily" should be "possibly"
We have made this change.

Referee #2:

In this manuscript Fréville et al present data on the morphological transformations of *P. falciparum* parasites immediately after they invade a new red blood cell (RBC). Very little research has so far focused on this life cycle phase and this manuscript is therefore of high interest and novelty. One part of this study describes the morphological alterations in WT parasites. The other part of the manuscript uses a conditional mutant of the gene encoding PV6, a lipid transfer protein. The mutant parasites invade RBCs, but do not make amöboid cells and have a smaller surface area than normal rings. The authors use these cells to test what happens if the transformation to the amöboid parasite does not occur. Interestingly they find evidence that the PV6 mutant is more prone to splenic clearance. A further interesting finding is that the PV6 mutant does not seem to export proteins.

This is a very interesting paper on a so far poorly studied part of the life cycle of the parasite. The data is clearly presented and I have only non-critical comments:

- The introduction states that PV6 mediates changes to the RBC that are needed for survival. Given the lack of export of RESA and possibly other exported proteins it is likely that this is due to a lack of protein export. Or is the idea that PV6 directly changes the RBC?

This is a very good point. We have added a sentence to mention the possibility that the effect of the parasite on the erythrocyte surface area is mediated by exported proteins p19, lines 23-28

- Fig. 2C - E. Did the PV6 mutant parasites move around in the RBC like is known from WT rings? The EMs in the manuscript indicate the mutant cells may be stuck in the periphery of the RBC. Could this contribute to the failure of proper development into a spherical ring? Determining whether the mutant is arrested in position or not would be easy (or probably was already done along the experiments shown in Fig. 2C-E) and might give further indications what is wrong with the PV6 mutant.

This is a valuable comment. We have included movies of wild type and mutant parasites less than two hours after invasion (Supp movies 3-6 in revised manuscript). These show that parasites lacking PV6 do not move to any large degree, but that they do appear to wiggle and that they are not attached to the erythrocyte plasma membrane. (p10, lines 3-6).

- L1-6/P8: the argument makes sense that PV6 is needed to increase surface area but not necessarily to develop an amöboid shape. However, the last sentence of this section is then somewhat confusing as it states PV6 is needed to progress into the amöboid form. Maybe it might be more accurate to turn this sentence around so that it states that PV6 is needed for the parasite to increase in surface area and that this in turn is a prerequisite to adopt an amöboid shape.

This is a very good suggestion, which we have incorporated, see p10, lines 10-13.

- same section and also other parts in the paper: 'PV6 is required to progress to the amöboid form'. To me it sometimes seems to imply that the amöboid form is equivalent to a life cycle stage the parasite has to reach. However, amöboidity is not a stage the parasite has to progress to but a state a typical ring can (maybe facultatively) adopt. This is not to say an amöboid shape is not a good indicator that a cell has reached the typical ring stage (i.e. completed the metamorphosis), but it might be good to use wording that makes clear it is a state, rather than a stage. It is even possible some rings never show an amöboid shape and remain in the extended spherical shape.

This is a good suggestion. We have changed the wording and now refer to the amoeboid shape, rather than stage, throughout the manuscript.

- Given their tiny appearance in Giemsa smears I have to admit I find it surprising that the volume of the mutant is larger than that of the WT (Fig. 3D). The surface measurement was reproduced in Fig. 4B and C. It would be nice to have the volume data for these experiments as well.

We have added the volume data for the models in Supp Figure 4B (now EV Fig 3D) and a brief mention on p9, lines 30-31.

- Some support information for Fig. 6 would be nice. For instance, it was before mentioned (L31/P8) that the parasite preparations had some schizonts in them. Was the rate of schizonts in the samples that went into these experiments comparable between WT and PV6 mutant?

For all our experiments, a synchronized culture was split into two parts 30 hours after invasion, one part was treated with DMSO, the other with rapamycin. As shown in Appendix Fig 1, there is no difference in the development between the DMSO-treated and rapamycin-treated parasites at this stage of the lifecycle and the subsequent invasion is not affected. Hence, these cultures will have contained equal proportions of schizonts and rings.

For the experiment shown in Fig. 6A it might have been possible to distinguish stage of the cells, were for instance schizont-infected RBCs excluded?

The ring-stage parasites and the schizonts were differentiated using SYBR Green staining. We have included a figure showing a histogram of the SYBR Green staining and several representative images that were obtained in the schizont peak and the ring peak (Appendix Fig S4). These images clearly show that the rings and the schizonts were differentiated, allowing us to remove the schizonts from our analysis. This is now described in the Methods section (p24, lines 20-24).

Was the input and output of the microfiltration smeared? There is a FACS plot for the spleen experiment which is a good way to determine parasitemia, but a smear gives more accurate information what exactly went in and came out.

We agree that seeing the parasites directly always adds something. Although we do not have the original slides for this experiment, we have included representative images from an experiment performed in an identical manner in Appendix Fig S5.

- I am aware that there might be a confounding effect on the RBC, but maybe this is still worth testing: did the authors run the microfiltration experiment (Fig. 6C) also with jasplakinolide treated WT comparing to control and the mutants (plus/minus) jasplakinolide? This might show whether amöboidity is a contributor of the effect.

To address this comment, we have added a panel to EV7A that shows the results of a microfiltration experiment with two different parasites lines treated with jasplakinolide. The results indicate that these parasites pass through the column similarly to untreated parasites. It is not clear how flexible the jasplakinolide-treated parasites are, however. This is described in the text on p13, lines 9-14.

- The authors write that PV6 promotes the passage through the spleen. This sounds very active, maybe safer to say that if it is not there, spleen passage is reduced.

We have changed the wording to 'whose activity has a positive effect on the passage the passage of infected erythrocytes through the host's spleen' – p13, lines 21-22.

- The use of the semicolon in the paper was slightly confusing to me. I am not a native speaker, but maybe this could be revisited, for instance by replacing with a colon in some places.

We have replaced most of the semicolons in the text. We left the ones that separated text in parentheses.

- L12/P5 remove the bracket with ref in it.

We thank the reviewer for spotting this embarrassing mistake. We have corrected this.

- In the discussion it is stated that: the PV6 parasites are smaller. But if I correctly

understand they are smaller in surface area but larger in size (Fig. 3D)? It is important to make this distinction.

We have changed the wording, p14, line 20.

- L28/P11 insert space before This
We have made this change.

Referee #3:

Freville et al report on the development of the human malaria parasite *Plasmodium falciparum* immediately after invasion of the host erythrocyte and the role of the phospholipid transfer protein PV6 in this process. The authors demonstrate that *Plasmodium falciparum* malaria parasites undergo multiple morphological changes soon after erythrocyte invasion and implicate this in the parasites ability to avoid splenic clearance in the host. They further demonstrate that the protein PV6 plays a role in at least one part of this process, the formation of amoeboid like rings. This manuscript is well written with only minor spelling/grammatical errors as detailed below. Previously published literature on this topic has been referenced appropriately and in context with the results presented here. Some of the interpretations, or their explanations, are a bit challenging to support and the manuscript would benefit from some more thought into how this could best be done.

Major Comments:

Text states that WT parasites with jasplakinolide had shape affected but not area. Based on 2C jasplakinolide appears to differ from DMSO-treated control which is amoeboid and seems to be more spherical than Rapalog treated samples but based on 2E, it doesn't seem that shape was greatly affected compared to DMSO control.

For 2D, there is a NS difference between DMSO and Jasplakinolide, but a Sig difference between DMSO/Jas and Rapamycin treatment. This is interpreted as Jasp not affecting area.

For 2E, there is a NS difference between DMSO and Jasplakinolide, but a Sig difference between DMSO/Jas and Rapamycin treatment. This is interpreted as Jasp affecting the shape??? Should not this comparison, as for area, be a comparison with DMSO, where there is NS difference for Jasp?

Based on this data, I would interpret that Jasp treatment showed no significance difference in Area or Shape when compared to the DMSO control.

We thank the reviewer for bringing this to our attention. The results that were presented clearly do not make much sense. In response to this comment, we have re-analyzed our raw data and discovered that in the course of producing the figure, some data sets were exported incorrectly, leading to an error in Fig 2E. We have corrected the error, which revealed that there is, as expected, a significant difference in the shape of parasites treated with DMSO and parasites treated with jasplakinolide (Fig 2E). The data on the area are unaffected, indicating that the jasplakinolide-treated parasites change shape but not size.

Page 12, line 1: Based on the lower erythrocyte volume for rapamycin treated PV6 loss of function mutants it is hypothesised that this protein is involved in mitigating erythrocyte volume loss. Is there any evidence that loss of PV6 leads to an increase in parasitophorous vacuole membrane volume, since this is where some erythrocyte membrane is lost. Have these measurements been done? If not, where could the membrane have gone? In the discussion, the possibility that PV6 somehow sends erythrocyte membrane back to the cell surface is put forward, though the authors acknowledge that this seems like a difficult thing for PV6 to achieve when it is located in the PV. How might this work then. Could it be at the point where the tight junction closes off so that less erythrocyte membrane is taken in during PVM formation? Is it possible that loss of PV6 selects for invasion into small erythrocytes somehow?

This is a very valid point. We believe that the missing surface area is represented by the membranous whorls that are frequently detected in the mutant parasites. The inability to transfer the lipids in these membranes keeps the PVM small and takes away a source of lipids for the mitigation of the surface area loss. We don't know how the mitigation of the loss of surface area occurs but speculate that this may be mediated by exported proteins (which are not exported by the PV6 mutant). As PV6 is within the merozoite, it is difficult to envision a scenario in which the parasite induces a preference for small erythrocytes. We have added a short paragraph about this in the Discussion – p15, lines 18-24.

There seems to be two hypotheses put forward into why splenocyte models preferentially removed PV6 loss of function infected erythrocytes over those that had PV6. The first is the smaller size of the erythrocyte with loss of PV6, but it seems counter intuitive that a smaller cell in a bead-based model be preferentially removed.

Although indeed counterintuitive, the loss of surface area is associated with a decrease in the surface area-to-volume ratio, which is a key determinant of RBC deformability. The loss of surface area-to-volume ratio in artificially created spherocytes was previously shown to be associated with an increased RBC retention rate in both microfiltration and *ex vivo* perfusion of human spleens (Safeukui, *et al* 2012, PMID: 22510876, Safeukui *et al* 2013, PMID: 23555907).

Second is the PV6 expressing parasites can form amoeboid shapes that make them, and their erythrocytes, more flexible. Whereas loss of PV6 maintains the round merozoite state. Merozoites are much smaller than the erythrocytes they infect. Would they be big enough to impact on progression through the spleen? These models are put forward largely separately and not really discussed together. This makes it unclear which, or whether both, models are likely contributing the most. Linking these models in the discussion and discussing whether either would impact the likelihood of the other would be of benefit.

Splenic retention of an infected red blood cell may depend on the overall rigidity of the red blood cell or the parasite it contains. In the case of mature forms, due to cytoadherence and rigidity, the infected erythrocyte does not even reach the spleen or is retained there immediately (Buffet *et al Blood* 2011, PMID: 20852127; Duez *et al AAC* 2015, PMID: 25941228). Rings on the other hand, retention is more

associated with loss of surface/volume ratio and the greater the surface area lost the more they will be retained (Safeuki et al PLoS One 2013 PMID: 23555907). It is technically difficult to carry out in vitro experiments to separate the part strictly linked to the parasite or the red blood cell, but it is known that, for example, the main mechanism of parasite clearance during treatment of malaria with artemisinin derivatives is linked to mechanical retention of the dead parasite in the inter-endothelial clefts of the splenic red pulp before it is eliminated by the pitting phenomenon (Buffet *et al. Blood* 2011, PMID: 20852127; Ndour *et al. J Infect Dis* 2015, PMID: 25183768; Picot, Ndour *et al. Am J Hematol* 2015, PMID: 25641515; Wojnarsky *et al. J Infect Dis* 2019, PMID: 30877300). Here, increased parasite volume associated with no increase in membrane surface area appears to be the key parameter for enhanced retention.

In the discussion, in the new paragraph about the mitigation of the loss of surface area, we also address the two models and state that we think that the loss of surface area is the main cause of the removal of infected erythrocytes in the spleen and spleen model, see p15, lines 18-24.

There are a number of small text errors, some of which are highlighted below. Page 5, line 12: remove '(ref)'. Page 6, line 4: Egress an invasion are said to be precisely timed, but then this is said to happen 'approximately' 15 minutes after removal of Compound2/ML10. Suggest that the word precisely is removed as there is a probably range of several minutes with this method.

We have changed to wording from 'precisely' to 'very narrow time window' p6, lines 10-11.

Page 6, line 12: The difference in the rate of conversion to the amoeboid form between two experiments is posited to be from observing the cells. Is it also possible that differences in the starting late-stage parasite population and relative period on Compound2/ML10 during development could also impact downstream events?

We deem it very unlikely that there are important differences in the late-stage parasites in the two experiments, as the parasites were prepared similarly for the invasion experiment and the video experiment. In both experiments, the incubation in the presence of ML10 did not exceed three hours, during which the parasites do not appear to be affected by the inhibition of egress [Ressurreição et al *PLOS ONE* 2019, PMID: 32673324].

Page 6, lines 21 - 32: Less amoeboid forms in Pk parasites. This is an interesting comparison, but there's no statement of the significance or further discussion of this so from that perspective, this paragraph lacks some relevance.

This is a very good comment. We have added a brief discussion of our findings with *P. knowlesi* and the importance of the cup shapes and amoeboids in the discussion (p 14, line 3-6).

Page 11, line 28: gap between sentences.
We have made this change.

Page 10, line 17: change 'can occurs' to either 'can occur' or 'occur'.
We have made this change.

Page 13, line 5: 'further likely', perhaps change to either just 'further' or just 'likely'.
We have made this change.

Page 16, line 5: The *P. falciparum* culture method mentions selecting for rings two different ways (precoll, sorbitol). Can this be reworded.
We have made this change.

Page 23, lines 27 and 31: 'Error bars' instead of 'Errors bars'.
We have made this change.

Page 24, line 5: change 'fornation'.
We have made this change.

Page 24, lines 27 and 29: 'Error bars' instead of 'Errors bars'.
We have made this change.

Page 25, line 19: Change 'parasites' to 'parasite', assuming singular.
We have made this change.

Page 25, line 20 - 21: 'Live cell imaging of an infected erythrocyte stained with ceramide, the DNA dye SiR-Hoechst, the merge of the ceramide and SiR-Hoechst images and imaged using DIC'. Confusing sentence.
We have modified this sentence.

Page 26, lines 11 and 17: 'Errors bars' again.
We have made this change.

Figure 1B: panels on the right depicting amoeboid forms. The 25-minute image could show a clearer amoeboid form if one is available.

We have left this image, as we think that it does show the amoeboid shape of the parasite.

Dear Dr. van Ooij,

Thank you for the submission of your revised manuscript to our editorial offices. I have now received the reports from the three referees that were asked to re-evaluate the study, you will find below. As you will see, the referees now fully support the publication of the study in EMBO reports. However, all three referees have remaining concerns and suggestions to improve the study, I ask you to address in a final revised manuscript. Please also provide a final p-b-p-response addressing the remaining points of the referees,

Moreover, I have these editorial requests I also ask you to address:

- There are author name discrepancies: It is Aissatou Bailo Dialo in the manuscript, but Diallo Aissatou Bailo in the submission system; and Papa Alioune Ndour in the manuscript text and Alioune Ndour in the submission system. Please check. The names should be similar in the system and on the title page of the manuscript text file.
- We now use CRediT to specify the contributions of each author in the journal submission system. CRediT replaces the author contribution section. Please use the free text box to provide more detailed descriptions and do NOT provide your final manuscript text file with an author contributions section. See also our guide to authors: <https://www.embopress.org/page/journal/14693178/authorguide#authorshipguidelines>
- We updated our journal's competing interests policy in January 2022 and request authors to consider both actual and perceived competing interests. Please review the policy <https://www.embopress.org/competing-interests> and add a statement declaring your competing interests. Please name that section 'Disclosure and Competing Interests Statement' and add it after the author contributions section.
- Please add a Data availability section (DAS) to the manuscript. Here information on primary datasets produced in this study (e.g. RNA-seq, ChIP-seq, structural and array data) that are deposited in a public database should be provided. If no primary datasets have been deposited, please also state this in this section (e.g. 'No primary datasets have been generated and deposited').
- Please add up to 5 keywords to the manuscript and order the sections like this using these names:
Title page - Abstract - Keywords - Introduction - Results - Discussion - Methods - Data availability section (DAS) - Acknowledgements (including the funding information) - Disclosure and Competing Interests Statement - References - Figure legends - Expanded View Figure legends.
- Please fill in the header of the author checklist (ID, author and journal).
- The Expanded View format, which will be displayed in the main HTML of the paper in a collapsible format, has replaced the Supplementary information. Please follow the nomenclature Figure EV1, Figure EV2 etc. for your EV figures in the file names, legends and in the callouts.
- The movies should be named Movie EVx. Please update the file names, legends and the callouts to Movie EV1 - Movie EV9. The legends need to be removed from the manuscript text file and each should be provided as a readme.txt file, ZIPped together with its movie file and uploaded as one ZIPed folder per movie.
- Figure EV6 is a table. I would suggest to move this to the Appendix (see below).
- All additional Supplementary material should be supplied as a single pdf file labeled Appendix. The Appendix should have page numbers and needs to include a table of content on the first page (with page numbers) and legends for all content. Please follow the nomenclature Appendix Figure Sx, Appendix Table Sx etc. throughout the text, and also label the figures and tables according to this nomenclature. Please also remove all the text related to Appendix material from the main manuscript text file.
- Please include all the methods information into the main methods section. We do not allow Supplementary methods.
- Please also make sure that all figure panels are called out separately and sequentially, using the correct nomenclature (see above).
- Please check again that the number "n" for how many independent experiments were performed, their nature (biological versus technical replicates), the bars and error bars (e.g. SEM, SD) and the test used to calculate p-values is indicated in the respective figure legends. Please also check that all the p-values are explained in the legend, and that these fit to those shown in the figure. Please provide statistical testing where applicable. Please avoid the phrase 'independent experiment', but clearly state if these were biological or technical replicates. Please also indicate (e.g. with n.s.) if testing was performed, but the differences are not significant. In case n=2, please show the data as separate datapoints without error bars and statistics. See

also:

<http://www.embopress.org/page/journal/14693178/authorguide#statisticalanalysis>

If $n < 5$, please show single datapoints for diagrams. There are presently diagrams that seem to miss the statistics, 'n.s.' or have only partial statistics (e.g. 1B, 2B, EV5B or EV7A/B). Please check. Moreover:

- Please note that the box plots need to be defined in terms of minima, maxima, centre, and percentile in the legend of figure 2B.
- Please provide the information related to n in the legends of figures EV3 C, D, EV5 B.
- Please note that $n=2$ in figure 1B.
- Please define the error bars in the legends of figures 3C, D; EV5 B.
- Please note that the measure of center for the error bars needs to be defined in the legends of figures 1F, 2D, E; 6C, EV3 C, D; EV7 A.

- Please provide the exact p-values in the legends of figures 1F, 2B, D, E; 3C, D; 6B, C; EV3 C
- Please indicate the statistical test used for data analysis in the legends of figures EV3 D

- Please add scale bars of similar style and thickness to microscopic images, using clearly visible black or white bars (depending on the background). Please place these in the lower right corner of the images themselves. Please do not write on or near the bars in the image but define the size in the respective figure legend. Presently, most of the scale bars are too thin or too small.

- Please make sure that all the funding information is also entered into the online submission system and that it is complete and similar to the one in the acknowledgement section of the manuscript text file. Presently, the funding from the University of Paris City is missing in the submission system.

- Please move the ethics statement to the methods section.

- Please add a paragraph titled 'Biosafety' to the methods section gathering all information on where and how biosafety-relevant experiments with pathogens were performed and that these were approved, and by whom (institution, government).

- Please remove the template text from the reagents and tools table and add callouts to the table in the Methods section where appropriate.

- There is an additional figure uploaded (Additional data for Reviewers only). Should this be displayed in the final manuscript? If yes, I would suggest to add this to one of the EV figures or to the Appendix.

- We noted a reuse of panels between Figure 3 A/B and x. The legend of Figure EV2 states: 'Three-dimensional models from SBF-SEM data of the parasites presented in Figure 2.' But the reuse is between Figure EV2 and Figure 3. Please check and correct the legend.

- Thank you for providing the requested source data. Please upload the source data for the main figures as one folder per figure (with all files for one figure in one folder and ZIPed). Inside each folder, the files should be organized in subfolders, one subfolder for each panel.

In addition, I would need from you uploaded separately:

Best,

Referee #1:

The authors have addressed my comments resulting in an improved manuscript. I have only a few minor comments regarding

the lack of statistical tests in a few places and lack of clarity in the description of the RESA export assay.

-P13, line 13: "assume" should be "to assume"

-P22, line 11-17: the methods description of how RESA export was quantified is difficult to understand. The statement "...regions of RESA expression inside and outside the red blood cell." is confusing as RESA associates with the RBC cytoskeleton and does not localize outside the RBC. Perhaps "outside the RBC" is meant to indicate the fraction of RESA that is not exported (not localized to the RBC periphery)? If so, a different term should be used since non-exported RESA is still inside the RBC. I'm also confused by this sentence: "The area measurement tool was then employed to quantify the proportion of EXP2 expression within and outside the red blood cell." Area measurements do not enable the amounts of EXP2 to be determined (area and fluorescence intensity are different measurements) and the concept of EXP2 outside the RBC doesn't make sense (same comment as RESA localization above). I'm unclear what about EXP2 is being measured and also not sure why EXP2 is being measured. Perhaps the authors used the region marked by EXP2 to delineate a mask for the PVM/parasite so any RESA in this region could be separated from the exported fraction of RESA, but this is not clear from the description. Please revise for clarity.

-P30, line 23: "anti-RESA staining within and outside the red blood cell was measured." I'm confused about this statement similar to the comment above about the methods. RESA should not be outside the RBC. While the y-axis label of the graph "proportion of RESA exported to the RBC membrane" makes sense, this does not agree with the statement "outside the red blood cell" in the text and this should be adjusted to match the figure y-axis label.

-EV5B: A statistical test has not been applied to this data. This is needed to support the claim that RESA export is significantly decreased in the rapamycin treated parasites.

-EV7A: A statistical test has not been applied to this data. This is needed to support the claim that retention of jasplakinolide-treated parasites is not altered relative to the DMSO control.

Referee #2:

The authors appropriately and thoroughly addressed most reviewer comments in this revised manuscript. They for instance now show that jasplakinolide has no effect on column retention of infected red blood cells, indicating it is the size of the infected erythrocyte that likely is the reason for the PV6-KO retention phenotype. They for instance also quantified whorls in the invaded rings which might give some explanation about the mechanism of PV6 function.

The one point that might need some further consideration is the conclusion in the revision that the PV6 mutant rings move away from the erythrocyte membrane (appendix movies 3-6). While I agree with the authors that these parasites wiggle, and one actually appears to turn around its own axis, there does not seem to be a lot of evidence in these movies that suggests they really get away from the membrane. Rather they seem to remain in some sort of contact with the host cell. Even if there is the occasional mutant in the center of an erythrocyte, it might still be adherent to the top or bottom of the host cell. For this reason, I would recommend to be careful not to fully discard the possibility that the mutants might remain in some ways attached or adherent to the erythrocyte membrane and that this could contribute to the phenotype.

Related to this, the legend to AM3 and AM4 indicate that the reader should note the movement of the limbs of the rapa-treated parasites, but AM3 and AM4 both show a spherical parasite without any limbs. Are these the correct movies? AM6 legend indicates it is a DMSO (control) parasite where a slight move away from the erythrocyte membrane is seen and that the cell is of small circular shape. But AM6 shows a parasite that becomes amoeboid but seems to be well stuck in the middle of the erythrocyte. Again, is this the cell the legend is referring to? AM5 has the same legend text but shows a ring that also does not move much but is amoeboid, not small nor spherical. Can the authors clarify this? Could it be that the lower part of the legend of AM3 and 4 was swapped with that of AM5 and 6? Still, even then the "slight move away from the erythrocyte membrane" in the PV6 mutant may not be a permanent detachment.

One other point that might be relevant considering these movies: it seems that both, mutant and WT, do not move much in the erythrocyte at this stage over the observed time frame, is this representative? The two mutant parasites seem to be stuck in the periphery, the WT parasites in the center of the erythrocyte. It is only two cells each, so this may not be representative but maybe this is a general observation? While not essential for the manuscript, clarifying this might give some hints about the function of PV6 and would be useful for the description of the KO phenotype.

A few other minor things:

P4/L18 (track changes word file): "are highly deformable, erythrocytes", consider replacing comma with a full stop.

P9L30 (track changes word file): In response to reviewer 1, volume measurements of the second SBF-SEM experiment were carried out which revealed no difference to control in that replicate. The authors write in the rebuttal that this may be due to the lower level of synchronicity of the parasites in that experiment. In the manuscript with track changes they added and again deleted a sentence to that end: "although this difference was not detected in an experiment using less synchronized parasites."

Instead "however in this experiment, no volume increase was observed [EV3D]" was added a few lines further down. Given that it is unclear why this difference between the experiments exists, it might be useful to include the suspicion that it might be due to a lower degree of synchronisation with the sentence that was kept in the manuscript.

P11/L30 "The presence of whorls in wildtype parasites indicates that the presence of whorls may represent a brief intermediate stage and that PV6 transfers the lipids from the whorls to other membranes." Somewhat repetitive, consider shortening.

P15/L22 consider replacing semicolon with a full stop.

Referee #3:

-Fig 5B on page 53 (merged file) and the associated figure legend on page 30, line 30, does not provide an explanation of the statistical test used or a description of the error bars.

-Page 13, line12-13: missing word, '...rather than the inability TO assume....'

-Appendix movie 5 and 6: I have looked at these a couple of times and I don't see what is being described in the legend. The only circular shape appears towards the end, otherwise it seems mainly amoeboid, although this seems to be phase dependent as its sometimes there and sometimes not. I also don't see the slight movement that is obvious. I mention this in case these are the wrong videos or figure legends. Appendix movie 3 and 4, for example, show a small round parasite, but the figure legend says to watch for amoeboid. Is the mix up there? Suggest cross checking the others as well.

Dear Achim,

Thank you very much for the opportunity to revise our manuscript. We thank the reviewers for their insightful comments to you and the reviewers for the comments. We have now addressed all the comments, as described below. In addition, we have added one reference (Polino et al) that we had overlooked in previous versions.

- There are author name discrepancies: It is Aissatou Bailo Dialo in the manuscript, but Diallo Aissatou Bailo in the submission system; and Papa Alioune Ndour in the manuscript text and Alioune Ndour in the submission system. Please check. The names should be similar in the system and on the title page of the manuscript text file.

We have changed the name of Alioune Ndour to Papa Alioune Ndour in the manuscript so that it is the same in the manuscript and submission system but have trouble with Aissatou Bailo Diallo's name. The name should be as written in the manuscript (Diallo is their family name), but we cannot change the information in the submission system (where Diallo is listed as first name from another manuscript submission). We have reached out to this author to ask whether they can make the required change. We received a response confirming that Diallo is their last name, but they have not yet changed their name in the system.

- We now use CRediT to specify the contributions of each author in the journal submission system. CRediT replaces the author contribution section. Please use the free text box to provide more detailed descriptions and do NOT provide your final manuscript text file with an author contributions section. See also our guide to authors:

<https://www.embopress.org/page/journal/14693178/authorguide#authorshipguidelines>

We have removed this section from the text. All author credits are added on the submission page.

- We updated our journal's competing interests policy in January 2022 and request authors to consider both actual and perceived competing interests. Please review the policy <https://www.embopress.org/competing-interests> and add a statement declaring your competing interests. Please name that section 'Disclosure and Competing Interests Statement' and add it after the author contributions section.

We have reviewed the policy and added the 'Disclosure and Competing Interests Statement' section.

- Please add a Data availability section (DAS) to the manuscript. Here information on primary datasets produced in this study (e.g. RNA-seq, ChIP-seq, structural and array data) that are deposited in a public database should be provided. If no primary datasets have been deposited, please also state this in this section (e.g. 'No primary datasets have been generated and deposited').

We have added this section.

- Please add up to 5 keywords to the manuscript and order the sections like this using these names:

Title page - Abstract - Keywords - Introduction - Results - Discussion - Methods - Data availability section (DAS) - Acknowledgements (including the funding information) -

Disclosure and Competing Interests Statement - References - Figure legends - Expanded View Figure legends.

We have added four keywords.

- Please fill in the header of the author checklist (ID, author and journal).

Our apologies for overlooking this with the previous submission, we have filled in the header.

- The Expanded View format, which will be displayed in the main HTML of the paper in a collapsible format, has replaced the Supplementary information. Please follow the nomenclature Figure EV1, Figure EV2 etc. for your EV figures in the file names, legends and in the callouts.

We have changed the names of the expanded view figures to Figure EVx

- The movies should be named Movie EVx. Please update the file names, legends and the callouts to Movie EV1 - Movie EV9. The legends need to be removed from the manuscript text file and each should be provided as a readme.txt file, ZIPped together with its movie file and uploaded as one ZIPed folder per movie.

We have submitted the movies as described.

- Figure EV6 is a table. I would suggest to move this to the Appendix (see below).

We have changed Figure EV6 to Appendix Table S2.

- All additional Supplementary material should be supplied as a single pdf file labeled Appendix. The Appendix should have page numbers and needs to include a table of content on the first page (with page numbers) and legends for all content. Please follow the nomenclature Appendix Figure Sx, Appendix Table Sx etc. throughout the text, and also label the figures and tables according to this nomenclature. Please also remove all the text related to Appendix material from the main manuscript text file.

We have now submitted the supplementary material as one Appendix file and have removed the Appendix legends from the main text.

- Please include all the methods information into the main methods section. We do not allow Supplementary methods.

We have combined the methods sections.

- Please also make sure that all figure panels are called out separately and sequentially, using the correct nomenclature (see above).

We have checked the nomenclature throughout.

- Please check again that the number "n" for how many independent experiments were performed, their nature (biological versus technical replicates), the bars and error bars (e.g. SEM, SD) and the test used to calculate p-values is indicated in the respective figure legends. Please also check that all the p-values are explained in the legend, and that these fit to those shown in the figure. Please provide statistical testing where applicable. Please

avoid the phrase 'independent experiment', but clearly state if these were biological or technical replicates. Please also indicate (e.g. with n.s.) if testing was performed, but the differences are not significant. In case $n=2$, please show the data as separate datapoints without error bars and statistics. See also:

<http://www.embopress.org/page/journal/14693178/authorguide#statisticalanalysis>

If $n < 5$, please show single datapoints for diagrams. There are presently diagrams that seem to miss the statistics, 'n.s.' or have only partial statistics (e.g. 1B, 2B, EV5B or EV7A/B). Please check.

1B and 1F: We have added data points to the graph.

2B: All statistical information for Figure 2B is provided in Appendix Table S1. We have added information about the statistics for those comparisons that are most relevant to the conclusions in the manuscript in the figure, but readers have access to information about all statistics in the Table. Adding this to the figure would make the figure so busy that it will be more difficult to interpret.

EV5B: We have added statistical information in the figure and the figure legend.

EV7A: This is now Figure EV6A. We have added the statistical information to the figure and the legend.

EV7B: This is only 1 biological replicate, this mentioned that in the legend. It represents a control experiment that shows that the column retains fixed but not unfixed erythrocytes. No conclusions are drawn from this figure other than that the column can differentiate erythrocytes that can

Moreover:

- Please note that the box plots need to be defined in terms of minima, maxima, centre, and percentile in the legend of figure 2B.

We have added this information in the legend.

- Please provide the information related to n in the legends of figures EV3 C, D, EV5 B.

We have added information about the n in the legends.

- Please note that $n=2$ in figure 1B.

We have indicated this in the legends and have included dots to represent the individual experiments in the figure.

- Please define the error bars in the legends of figures 3C, D; EV5 B.

The error bars have been defined in the legends.

- Please note that the measure of center for the error bars needs to be defined in the legends of figures 1F, 2D, E; 6C, EV3 C, D; EV7 A.

1F: We have added this information to the legend and have replaced the figure with a version that shows the datapoints.

2D and E: We have added this information to the legend: Error bars +/- SD

6C: We have added this information to the legend: Error bars +/- SD

EV3C and D: We have added this information to the legend: Error bars +/- SD

EV7A: We have added this information to the legend: Error bars +/- SD

- Please provide the exact p-values in the legends of figures 1F, 2B, D, E; 3C, D; 6B, C; EV3C

1F: We have added the exact p-value to the figure legend

2B: All p values are listed in Appendix table S1. We feel that there are too many to list in the legend and that describing them all will add a lot of text to the legend.

2D: Our software (GraphPad Prism 10) does not provide us with a more accurate p value for this.

2E: Our software (GraphPad Prism 10) does not provide us with a more accurate p value for this.

3C, 3D, 6B, 6C and EV3C: We have added the exact p-values to the figure legends

- Please indicate the statistical test used for data analysis in the legends of figures EV3D

We have added this information to the figure legend.

- Please add scale bars of similar style and thickness to microscopic images, using clearly visible black or white bars (depending on the background). Please place these in the lower right corner of the images themselves. Please do not write on or near the bars in the image but define the size in the respective figure legend. Presently, most of the scale bars are too thin or too small.

We have doubled the thickness of the scale bars.

- Please make sure that all the funding information is also entered into the online submission system and that it is complete and similar to the one in the acknowledgement section of the manuscript text file. Presently, the funding from the University of Paris City is missing in the submission system.

We have updated the funding information in the submission system with the funding obtained from the University of Paris City.

- Please move the ethics statement to the methods section.

We have moved the ethics statement to the Methods section.

- Please add a paragraph titled 'Biosafety' to the methods section gathering all information on where and how biosafety-relevant experiments with pathogens were performed and that

these were approved, and by whom (institution, government).

We have added this section. In it we list where the genetic modifications of the parasites were performed and the organization that has approved the experiments.

- Please remove the template text from the reagents and tools table and add callouts to the table in the Methods section where appropriate.

We have added reagents to the Reagents and Tools table and have included this as Appendix Table S3. A call out to the table has been added to the text after the first mention of each reagent.

- There is an additional figure uploaded (Additional data for Reviewers only). Should this be displayed in the final manuscript? If yes, I would suggest to add this to one of the EV figures or to the Appendix.

This figure was meant only for the reviewers. It should not be included in the published version of the manuscript.

- We noted a reuse of panels between Figure 3 A/B and x. The legend of Figure EV2 states: 'Three-dimensional models from SBF-SEM data of the parasites presented in Figure 2.' But the reuse is between Figure EV2 and Figure 3. Please check and correct the legend.

We have corrected the legend.

- Thank you for providing the requested source data. Please upload the source data for the main figures as one folder per figure (with all files for one figure in one folder and ZIPed). Inside each folder, the files should be organized in subfolders, one subfolder for each panel.

We have zipped and uploaded all the files in the form requested.

In addition, I would need from you uploaded separately:

- a short, two-sentence summary of the manuscript (not more than 35 words).

We have added a short summary

- two to four short (!) bullet points highlighting the key findings of your study (two lines each).

We have added these bullet points

- a schematic summary figure as separate file that provides a sketch of the major findings (not a data image) in jpeg or tiff format (with the exact width of 550 pixels and a height of not more than 400 pixels) that can be used as a visual synopsis on our website.

We have included a summary figure

Please use this link to submit your revision: <https://embor.msubmit.net/cgi-bin/main.plex>

Best,

Referee #1:

The authors have addressed my comments resulting in an improved manuscript. I have only a few minor comments regarding the lack of statistical tests in a few places and lack of clarity in the description of the RESA export assay.

-P13, line 13: "assume" should be "to assume"

We have made this change (p 13, lines 19-20).

-P22, line 11-17: the methods description of how RESA export was quantified is difficult to understand. The statement "...regions of RESA expression inside and outside the red blood cell." is confusing as RESA associates with the RBC cytoskeleton and does not localize outside the RBC. Perhaps "outside the RBC" is meant to indicate the fraction of RESA that is not exported (not localized to the RBC periphery)? If so, a different term should be used since non-exported RESA is still inside the RBC. I'm also confused by this sentence: "The area measurement tool was then employed to quantify the proportion of EXP2 expression within and outside the red blood cell." Area measurements do not enable the amounts of EXP2 to be determined (area and fluorescence intensity are different measurements) and the concept of EXP2 outside the RBC doesn't make sense (same comment as RESA localization above). I'm unclear what about EXP2 is being measured and also not sure why EXP2 is being measured. Perhaps the authors used the region marked by EXP2 to delineate a mask for the PVM/parasite so any RESA in this region could be separated from the exported fraction of RESA, but this is not clear from the description. Please revise for clarity.

The methods have been modified accordingly.

-P30, line 23: "anti-RESA staining within and outside the red blood cell was measured." I'm confused about this statement similar to the comment above about the methods. RESA should not be outside the RBC. While the y-axis label of the graph "proportion of RESA exported to the RBC membrane" makes sense, this does not agree with the statement "outside the red blood cell" in the text and this should be adjusted to match the figure y-axis label.

The reviewer is correct and we have made changes in the text and figure legends to reflect the localization of the RESA at the erythrocyte periphery.

-EV5B: A statistical test has not been applied to this data. This is needed to support the claim that RESA export is significantly decreased in the rapamycin treated parasites.

We have modified the legend and describe that the data are from 18 parasites, obtained in two individual experiments. We have added the following mention of the statistical analysis: Error bars +/- SD. The Mann-Whitney U test was performed for statistical analysis (**P < 0.001 (P=0.0005)).

-EV7A: A statistical test has not been applied to this data. This is needed to support the claim that retention of jasplakinolide-treated parasites is not altered relative to the DMSO control.

These data represent 2 replicates, as indicated in the legend. We have added the statistical analysis in the figure. We have added the following mention of the statistical analysis: Error bars +/- SD. The Mann–Whitney U test was performed for statistical analysis (** * *P < 0.0001).

Referee #2:

The authors appropriately and thoroughly addressed most reviewer comments in this revised manuscript. They for instance now show that jasplakinolide has no effect on column retention of infected red blood cells, indicating it is the size of the infected erythrocyte that likely is the reason for the PV6-KO retention phenotype. They for instance also quantified whorls in the invaded rings which might give some explanation about the mechanism of PV6 function.

The one point that might need some further consideration is the conclusion in the revision that the PV6 mutant rings move away from the erythrocyte membrane (appendix movies 3-6). While I agree with the authors that these parasites wiggle, and one actually appears to turn around its own axis, there does not seem to be a lot of evidence in these movies that suggests they really get away from the membrane. Rather they seem to remain in some sort of contact with the host cell. Even if there is the occasional mutant in the center of an erythrocyte, it might still be adherent to the top or bottom of the host cell. For this reason, I would recommend to be careful not to fully discard the possibility that the mutants might remain in some ways attached or adherent to the erythrocyte membrane and that this could contribute to the phenotype.

We have altered the text to indicate that it is possible that the PV containing a parasite lacking PV6 does not fully disengage from the erythrocyte membrane (page 8, lines 20-22).

Related to this, the legend to AM3 and AM4 indicate that the reader should note the movement of the limbs of the rapa-treated parasites, but AM3 and AM4 both show a spherical parasite without any limbs. Are these the correct movies? AM6 legend indicates it is a DMSO (control) parasite where a slight move away from the erythrocyte membrane is seen and that the cell is of small circular shape. But AM6 shows a parasite that becomes amoeboid but seems to be well stuck in the middle of the erythrocyte. Again, is this the cell the legend is referring to? AM5 has the same legend text but shows a ring that also does not move much but is amoeboid, not small nor spherical. Can the authors clarify this? Could it be that the lower part of the legend of AM3 and 4 was swapped with that of AM5 and 6? Still, even then the "slight move away from the erythrocyte membrane" in the PV6 mutant may not be a permanent detachment.

We did indeed switch the descriptions of the movies in the legends. This has now been corrected.

One other point that might be relevant considering these movies: it seems that both, mutant and WT, do not move much in the erythrocyte at this stage over the observed time frame, is

this representative? The two mutant parasites seem to be stuck in the periphery, the WT parasites in the center of the erythrocyte. It is only two cells each, so this may not be representative but maybe this is a general observation? While not essential for the manuscript, clarifying this might give some hints about the function of PV6 and would be useful for the description of the KO phenotype.

This is an interesting point and we have added text (p 8, lines 23-24) to make this point.

A few other minor things:

P4/L18 (track changes word file): "are highly deformable, erythrocytes", consider replacing comma with a full stop.

We have made this change.

P9L30 (track changes word file): In response to reviewer 1, volume measurements of the second SBF-SEM experiment were carried out which revealed no difference to control in that replicate. The authors write in the rebuttal that this may be due to the lower level of synchronicity of the parasites in that experiment. In the manuscript with track changes they added and again deleted a sentence to that end: "although this difference was not detected in an experiment using less synchronized parasites." Instead "however in this experiment, no volume increase was observed [EV3D]" was added a few lines further down. Given that it is unclear why this difference between the experiments exists, it might be useful to include the suspicion that it might be due to a lower degree of synchronisation with the sentence that was kept in the manuscript.

We have added a line in the manuscript to reflect this (p 10, lines 11-12).

P11/L30 "The presence of whorls in wildtype parasites indicates that the presence of whorls may represent a brief intermediate stage and that PV6 transfers the lipids from the whorls to other membranes." Somewhat repetitive, consider shortening.

We have rephrased this sentence to increase the clarity (p 12, lines 4-6).

P15/L22 consider replacing semicolon with a full stop.

We have made this change (p 15, lines 22).

Referee #3:

-Fig 5B on page 53 (merged file) and the associated figure legend on page 30, line 30, does not provide an explanation of the statistical test used or a description of the error bars.

We presume that the reviewer refers to EV5B, the quantitation of the export of RESA. Although these data are from two experiments (as indicated in the legend), we have added statistical analysis.

-Page 13, line12-13: missing word, '...rather than the inability TO assume...'

We have made this change (p 13, lines 27-28).

-Appendix movie 5 and 6: I have looked at these a couple of times and I don't see what is being described in the legend. The only circular shape appears towards the end, otherwise it seems mainly amoeboid, although this seems to be phase dependent as its sometimes there and sometimes not. I also don't see the slight movement that is obvious. I mention this in case these are the wrong videos or figure legends. Appendix movie 3 and 4, for example, show a small round parasite, but the figure legend says to watch for amoeboid. Is the mix up there? Suggest cross checking the others as well.

We had swapped the description of the parasites in the legends of the movies in the previous submission, leading to the confusion of the reviewer. We have corrected the legends for Movies EV3-6.

Dr. Christiaan van Ooij
The Francis Crick Institute
United Kingdom

Dear Dr. van Ooij,

Thank you for the submission of your final revised manuscript to our editorial offices. I now looked through the revised manuscript and your final p-b-p-response and consider the remaining points of the referees and the editorial requests as adequately addressed.

I am thus very pleased to accept your manuscript for publication in the next available issue of EMBO reports. Thank you for your contribution to our journal.

Yours sincerely,
